# CONTROLLED TEXT GENERATION VIA LANGUAGE MODEL ARITHMETIC

**Jasper Dekoninck, Marc Fischer, Luca Beurer-Kellner, Martin Vechev**
Department of Computer Science
ETH Zurich, Switzerland
`{jasper.dekoninck,marc.fischer,luca.beurer-kellner,martin.vechev}@inf.ethz.ch`

## ABSTRACT

As Large Language Models (LLMs) are deployed more widely, customization with respect to vocabulary, style, and character becomes more important. In this work, we introduce model arithmetic, a novel inference framework for composing and biasing LLMs without the need for model (re)training or highly specific datasets. In addition, the framework allows for more precise control of generated text than direct prompting and prior controlled text generation (CTG) techniques. Using model arithmetic, we can express prior CTG techniques as simple formulas and naturally extend them to new and more effective formulations. Further, we show that speculative sampling, a technique for efficient LLM sampling, extends to our setting. This enables highly efficient text generation with multiple composed models with only marginal overhead over a single model. Our empirical evaluation demonstrates that model arithmetic allows fine-grained control of generated text while outperforming state-of-the-art on the task of toxicity reduction. We release an open source easy-to-use implementation of our framework at `https://github.com/eth-sri/language-model-arithmetic`.

## 1 INTRODUCTION

In recent years, Large Language Models (LLMs) (Brown et al., 2020; Chowdhery et al., 2022; Touvron et al., 2023) have been increasingly recognized for their capabilities in handling a wide range of tasks (Rozière et al., 2023; Ouyang et al., 2022). In many applications, such as chatbots interacting with diverse audiences like children, students, or customers, precise control and customization of attributes such as the employed vocabulary, linguistic style, and emotional expression are crucial.

**Controlling Language Models**   A common technique for this is prompting with natural language (Ouyang et al., 2022). While prompting is simple and makes it easy to condition the LLM to a broad attribute, the ambiguity of natural language makes it challenging to express how present that attribute should be in the generated text. Further, prompting also lacks the ability to effectively steer the model away from a certain attribute in a reliable manner, as mentioning a specific topic in the prompt can inadvertently increase the likelihood of the model generating text about it (Jang et al., 2022), e.g. "do not mention cats" may increase the likelihood of the model referring to cats. One alternative is fine-tuning the model, but this requires highly specific training data for the desired attribute, which also has to implicitly encode the strength of the conditioning. Controlled Text Generation (CTG) techniques aim to solve this problem by steering the model during inference instead (Liu et al., 2021; Dathathri et al., 2020; Yang and Klein, 2021): The model is conditioned on a particular attribute $a$ in a smoothly controllable way, by biasing the model's token distribution. Many CTG methods are inspired by Bayes rule $P(\text{text}|a) \propto P(a|\text{text})P(\text{text})$, and utilize an auxiliary model, i.e. $P(a|\text{text})$, to condition the LLM, i.e., $P(\text{text})$, towards $a$.

**Key Challenge: Lack of Expressive and Efficient Control for Text Generation**   These techniques, however, suffer from several drawbacks, including a lack of expressiveness, efficiency, and interpretability. First, to control the strength of the applied conditioning, a parameter $\lambda$ is introduced in an ad-hoc manner, i.e., as an exponential weight $P(a|\text{text})^\lambda$. However, introducing the strength in this way, while possible, quickly becomes unintuitive as it can no longer be interpreted in a Bayesian

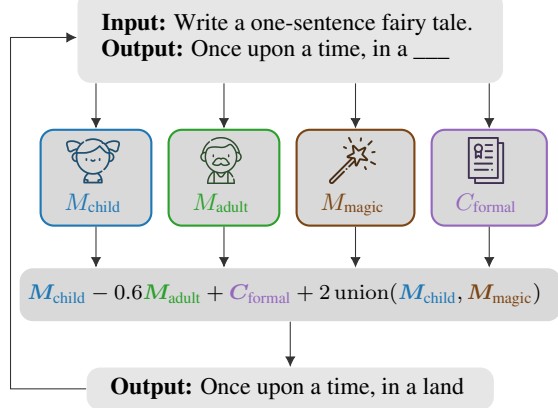

**Write a one-sentence fairy tale.**

$M_{\text{child}}$
Once upon a time, in a magical kingdom far, far away, there lived a brave and curious little princess who rode a sparkly unicorn and outwitted a grumpy dragon

$M_{\text{child}} - 0.6 M_{\text{adult}}$
Once upon a time, there was a sparkly rainbow unicorn that pooped ice cream and took me on a magical adventure to a land made entirely of candy!

$M_{\text{child}} - 0.6 M_{\text{adult}} + C_{\text{formal}}$
Once upon a time, in a magical land full of fluffy clouds and sparkly rainbows, there lived a little baby unicorn named Sparkles who had a beautiful, shimmering mane

$M_{\text{child}} - 0.6 M_{\text{adult}} + C_{\text{formal}} + 2 \, \text{union}(M_{\text{child}}, M_{\text{magic}})$
Once upon a time, in a land far, far away, there was a magical kingdom filled with sparkly unicorns, fluffy dragons, and a princess who could make ice cream appear out of thin air!

Figure 1: Overview of model arithmetic using an illustrative example. We outline the procedure for generating a fairy tale (left) using the models $M_{\text{child}}$, $M_{\text{adult}}$, $M_{\text{magic}}$ that produce text conditioned on the attributes *child*, *adult*, and *magic*, respectively and $C_{\text{formal}}$ a classifier for the formality of text. The right table shows example outputs for different (partial) formulas. Image attribution in App. C.

manner, e.g., when biasing away from attributes. Moreover, neither prompting nor CTG methods allow for the natural and controlled combination of multiple attributes or instructions with relative strength. This is due to the inherent ambiguity of natural language in prompting (Arora et al., 2023; Zhao et al., 2021), and the absence of a theoretical foundation and intuitive semantics for the biasing strength $\lambda$ with CTG methods. Lastly, both CTG techniques and fine-tuning often require custom and highly specific training data for the desired attribute (Yang and Klein, 2021; Sansone and Manhaeve, 2022; Saha et al., 2022; Kim et al., 2023) and can be resource-intensive (Kumar et al., 2022; 2021) as multiple models are evaluated at inference time.

**Fine-Grained Control via Model Arithmetic**  In this work, we address these challenges and introduce *model arithmetic*, a principled and intuitive method to combine multiple models. Our method is orthogonal to prompting, fine-tuning, and simple CTG concepts, like the use of classifiers, and can naturally incorporate them. Model arithmetic enables us to blend multiple LLMs and attributes into a single precisely controlled, formula-based composite model. To illustrate our method, consider the simple example in Fig. 1, where we aim to write a magical, child-like fairy tale. We employ multiple models $M_a$, with different attributes $a$. On the top right, we see a prompted model $M_{\text{child}}$ that already generates a child-appropriate story. However, the resulting text is not child-like and we therefore subtract an adult-conditioned model, $M_{\text{adult}}$, with a weight of $0.6$ to generate a less adult-sounding story. Now, to again increase formality, we additionally bias with classifier $C_{\text{formal}}$. Lastly, we use a special union operator to obtain a model that emphasizes both magical and child-like language and use it to further bias generation and obtain our final result. This simple example cannot be precisely expressed with prior CTG approaches and showcases the flexibility of model arithmetic. That is, it allows us to compose models in a natural way, while precisely controlling the impact of each component. Further, we can naturally incorporate paradigms such as prompting or fine-tuning (for the individual $M$ and $C$) and even implement many prior CTG techniques (discussed in §3) as simple formulas.

**Efficient Model Arithmetic via Generalized Speculative Sampling**  CTG methods, including model arithmetic, can lead to increased inference times as multiple models need to be evaluated in order to generate text. To counteract this, we generalize speculative sampling (Chen et al., 2023) to model arithmetic. Speculative sampling is usually employed to reduce the latency of a single LLM by augmenting it with a smaller model that proposes tokens, which are then validated by the LLM. In contrast, we extend it in a way where we postpone the evaluation of more expensive model calls within model arithmetic formulas. This allows us to execute model formulas comprised of multiple models with only marginal overhead over a single model and reduces model calls by up to $64\%$. The resulting inference speedup naturally extends to prior CTG techniques that can be expressed in model arithmetic (Pei et al., 2023; Sanchez et al., 2023; Chen et al., 2022; Schick et al., 2021).

**Key Contributions**    Our core contributions include:

- Model Arithmetic: A principled framework for fine-grained CTG, enabling precise control over multiple attributes. Our framework can express many prior CTG approaches (§3).

- An extension of speculative sampling to model arithmetic, counteracting the overhead of CTG and enabling efficient inference, which naturally benefits CTG techniques expressible in model arithmetic (§4).

- An extensive qualitative and quantitative evaluation of model arithmetic (§5). We show that it is more expressive than prior CTG work and outperforms them in toxicity reduction. We demonstrate that our extended speculative sampling reduces model calls by up to $64\%$.

## 2    BACKGROUND

We briefly introduce the required background and notation used in the remainder of the paper.

**Discrete Probability Distributions**    A discrete probability distribution $P$ associates a probability $P(x)$ with every element $x$ in a finite set $T$. For language modeling, this finite set is usually a set of tokens (or subwords). We often want to compute the probability of a token $x_k$ given all previous tokens $x_1, ..., x_{k-1}$ in a sequence, which we denote as $P(x_k|x_{1:k-1})$. We use the Kullback-Leibler (KL) divergence to measure the similarity of two distributions $P$ and $Q$:

$$D_{\mathrm{KL}}(P||Q|x_{1:k-1}) = \sum_{x \in T} P(x|x_{1:k-1}) \log \frac{P(x|x_{1:k-1})}{Q(x|x_{1:k-1})},$$

where we append $|x_{1:k-1}$ to denote conditioning on a sequence of tokens $x_{1:k-1}$. If this is implied by the context, we will omit the conditioning on $x_{1:k-1}$ and simply write $D_{\mathrm{KL}}(P||Q)$.

**Autoregressive Large Language Models**    Large Language Models (LLMs) are trained to generate sequences of tokens. Most recently, successful LLMs are autoregressive, i.e., they generate output token-by-token by modeling the probability distribution $P(x_k|x_{1:k-1})$ and sampling one token at a time from that distribution. Whenever we refer to a language model $M$, we directly refer to this distribution and denote it as $M(x_k|x_{1:k-1})$.

**Controlled Text Generation**    As introduced in §1, CTG techniques aim to introduce a given attribute $a$ (e.g. style or topic) in the output of a language model $M$, by biasing its distribution with respect to $a$. Oftentimes, a strength parameter $\lambda$ controls the strength of this conditioning. The conditioning model $P(a|\text{text})$ is modeled with a classifier (Yang and Klein, 2021; Sitdikov et al., 2022; Kim et al., 2023; Sansone and Manhaeve, 2022; Saha et al., 2022), a smaller finetuned model (Liu et al., 2021), or with the same model $M$ using a different prompt (Pei et al., 2023; Schick et al., 2021; Sanchez et al., 2023; Chen et al., 2022). In the first two cases, the biasing models have to be trained ahead of time. Many of these approaches are based on (a variant of) Bayes rule (Sanchez et al., 2023; Liu et al., 2021; Pei et al., 2023; Hallinan et al., 2023; Yang and Klein, 2021).

**Speculative Sampling**    Speculative sampling (Chen et al., 2023) speeds up inference of autoregressive language models by using a small proposal model $m$ to generate several tokens $x_1, ..., x_k$ and then validates these tokens using a bigger, more capable model $M$. Due to the way the underlying transformer architecture of current LLMs (Vaswani et al., 2017) works, this validation call is significantly cheaper than generating the tokens with $M$ directly.

Specifically, the entire sequence of proposed tokens $x_1, ..., x_k$ can be validated by a single, retroactive call to $M$. If token $x_i$ is rejected by $M$, all subsequent tokens $x_{i+1}, ..., x_k$ are discarded and $x_i$ is resampled. If all tokens are accepted, the next token $x_{k+1}$ can be directly sampled using the result of the same validation call to $M$. Thus, one can generate up to $k+1$ tokens with just a single call to $M$. Importantly, this procedure of accepting and resampling tokens ensures that the resulting distribution is equivalent to drawing token samples directly from $M$. For reference, we include the full speculative sampling procedure in Algorithm 1 of App. E.1.

## 3    Model Arithmetic

In this section we introduce model arithmetic, a principled approach for advanced CTG that enables the precise composition of language models, resulting in a distribution $P$ that can be sampled like a language model. This addresses the previously discussed drawbacks of prior CTG methods.

To this end, we first outline how an *output* distribution $P$ is constructed from a set of *input* distributions $Q_1, \ldots, Q_n$, by minimizing a linear combination of (weighted) KL-divergences $D_{\mathrm{KL}}(P||Q_i)$. Then we show how model arithmetic can be used to describe these distributions in a natural way. We defer all proofs to App. D.

**(Weighted) KL-Optimality**   The standard KL-divergence $D_{\mathrm{KL}}(P||Q)$ attends to each token by an equal amount, which might not always be desirable in the CTG setting. Indeed, suppose $Q$ represents the distribution of a certain attribute $a$ that we want to introduce in the output distribution $P$. When certain tokens are generally more associated with the attribute $a$, we might give the term $D_{\mathrm{KL}}(P||Q)$ more weight for these tokens, allowing to more strongly bias these specific tokens while reducing the bias for less important tokens. We therefore introduce the *weighted KL-divergence* $D_{\mathrm{KL}}^{[f]}$ as

$$D_{\mathrm{KL}}^{[f]}(P||Q|x_{1:k-1}) = \sum_x P(x|x_{1:k-1})f(x, x_{1:k-1}) \log \frac{P(x|x_{1:k-1})}{Q(x|x_{1:k-1})}$$

where $f\colon T \times T^{k-1} \to \mathbb{R}$ assigns a weight to each token $x \in T$, conditioned on $x_{1:k-1}$. We will later show how high-level constructs in model arithmetic map to particular choices of $f$.

Theorem 1 now defines and solves the problem of combining arbitrary probability distributions into a single output distribution by framing it as a minimization problem over a linear combination of weighted KL-divergences.

**Theorem 1** (Weighted KL-Optimality). *Let $T$ be the set of all tokens and $x_1, ..., x_{k-1}$ be a sequence of tokens such that $x_i \in T$. Then, given distributions $Q_1, \ldots, Q_n$ over $T$, functions $f_1, \ldots, f_n\colon T \times T^{k-1} \to \mathbb{R}$, and under mild technical assumptions detailed in App. D.1, the solution to the optimization problem for the generation of token $x_k$*

$$\arg\min_P \sum_{i=1}^n D_{\mathrm{KL}}^{[f_i]}(P||Q_i|x_{1:k-1}) \tag{1}$$

*is given by*

$$P(x_k = x|x_{1:k-1}) = \sigma \left( \frac{1}{\sum_{i=1}^n f_i(x, x_{1:k-1})} \sum_{i=1}^n f_i(x, x_{1:k-1}) \log Q_i(x|x_{1:k-1}) \right) \tag{2}$$

*where $\sigma$ is the softmax function.*

We note that this result applies more broadly than the autoregressive setting. Instead of conditioning on $x_{1:k-1}$, one can condition on $x_{1:k-1}, x_{k+1:t}$ without otherwise modifying the theorem.

Further, we can write $f_i(x, x_{1:k-1}) = \lambda_i(x_{1:k-1})f_i'(x, x_{1:k-1})$ where we factor $f_i$ into a part $\lambda_i$ that only depends on the context (i.e., the previous tokens) for scaling, and $f_i'(x, x_{1:k-1})$ that encodes token specific weights.

**Model Arithmetic**   Distribution $P$, resulting from Eq. (2), is completely determined by $\lambda_i(x_{1:k-1})$, $f_i'(x, x_{1:k-1})$ and $\log Q_i(x|x_{1:k-1})$ for all $x \in T$ and $i \in \{1, \ldots, n\}$. Since $T$ is a finite set, we can write $f_i'(x, x_{1:k-1})$ and $\log Q_i(x|x_{1:k-1})$ as vectors $\boldsymbol{f_i'} := (f_i'(x, x_{1:k-1}))_{x \in T}$ and $\boldsymbol{Q_i} := (\log Q_i(x|x_{1:k-1}))_{x \in T}$. Finally, since the normalization is completely determined by $\lambda_i$ and $\boldsymbol{f_i'}$, we can drop this in our notation and write $F = \sum_{i=1}^n \lambda_i \boldsymbol{f_i'} \boldsymbol{Q_i}$, where vector multiplication is element-wise. We drop $\lambda_i$ and $\boldsymbol{f_i'}$ when they are 1.

This notation makes it possible to use simple arithmetic operations to combine different input prompts, attributes, language models, and classifiers. We thus call this notation *model arithmetic*. Next, we discuss the operators in model arithmetic along with motivating examples (further shown in App. I) and summarize this in Table 1.

Table 1: Overview of *Model Arithmetic* where $\mathcal{I}_1(x) := [Q_1(x) > Q_2(x)]$ and $\mathcal{I}_2(x) := 1 - \mathcal{I}_1(x)$, $C$ is a classifier and $U$ the uniform distribution.

| | Model Arithmetic | Optimization Problem |
|---|---|---|
| Linear Combination | $\sum_i \lambda_i \boldsymbol{Q_i}$ | $\sum_i \lambda_i D_{\mathrm{KL}}(P\|Q_i)$ |
| Classifier | $\lambda \boldsymbol{C}$ | $\lambda\left(D_{\mathrm{KL}}(P\|Q_C) - D_{\mathrm{KL}}(P\|U)\right)$ |
| Union | $\mathrm{union}(\boldsymbol{Q_1}, \boldsymbol{Q_2})$ | $D_{\mathrm{KL}}^{[\mathcal{I}_1]}(P\|Q_1) + D_{\mathrm{KL}}^{[\mathcal{I}_2]}(P\|Q_2)$ |

Table 2: Prompt arithmetic examples using Llama-2-Chat-13b.

| **Tell me something interesting about pandas.** |
|---|
| $\boldsymbol{M}_{\text{formal}}$ |
| Certainly! Pandas are fascinating creatures, known for their distinct black and white markings … |
| $2\boldsymbol{M}_{\text{formal}} - \boldsymbol{M}$ |
| Certainly, user. The giant panda, scientifically known as Ailuropoda melanoleuca, is a intriguing and unique species of bear … |

**Linear Combinations** Many useful properties can be expressed as a linear combination of probability distributions $\sum_{i=1}^{n} \lambda_i \boldsymbol{Q_i}$, with $\lambda_i \in \mathbb{R}$. Most commonly, linear formulas include the standard output of an LLM $M$ as $Q_1$ (with $\lambda_1 = 1$) and additional distributions $Q_i$ are then used to bias the overall output towards (if $\lambda_i > 0$) or away from (if $\lambda_i < 0$) a certain attribute.

This can be used to combine several characteristics into a single persona, for model ensembling, and can also express prior CTG approaches (Liu et al., 2021; Pei et al., 2023; Sanchez et al., 2023; Chen et al., 2022) as shown in App. A. Table 2 show the results of linearly composing a non-conditioned model and a prompted *formal* model using a negative coefficient. As shown, the resulting composite model generates much more formal output than with standard prompting $\boldsymbol{M}_{\text{formal}}$.

**Classifiers** Binary classifiers that associate a probability with an input text can also be used to guide the output distribution towards the classified attribute (cf. Yang and Klein (2021); Sitdikov et al. (2022)). These classifiers can express attributes that are not easily expressible in natural language, such as the reward model in RLHF (Ouyang et al., 2022) or detection of AI-generated text (Solaiman et al., 2019) as shown in Table 3. There, we generate text that resembles human content more closely by using a classifier that detects AI-generated text (Solaiman et al., 2019) and bias away from it.

Table 3: Example using the GPT2-XL model and a detector $\boldsymbol{C}_{\text{gpt2-detector}}$ for it.

| **I like to** |
|---|
| $\boldsymbol{M}_{\text{gpt2}}$ |
| think of myself as a pretty good cook. I've made a lot of food, and I've learned a lot about cooking. I've also learned a lot about the world of food, and the people who eat it. |
| $\boldsymbol{M}_{\text{gpt2}} - 4\boldsymbol{C}_{\text{gpt2-detector}}$ |
| believe that I'm a pretty good judge of character. I watch a lot of TV - I'm a big fan of The Walking Dead, Game of Thrones and The Big Bang Theory … |

Classifiers do not output a token-level probability distribution and therefore do not permit the direct application of Theorem 1. However, to let a binary classifier $C \colon T^n \to [0, 1]$ guide the output distribution, we would want to minimize (or maximize) the expected cross-entropy of the classifier. Given $x_{1:k-1}$, the expected cross-entropy for $x_k$ under $P$ for the next token is given by

$$\mathbb{E}_{x_k \sim P}[-\log C(x_{1:k})] = -\sum_{x_k \in T} P(x_k|x_{1:k-1}) \log(C(x_{1:k})). \tag{3}$$

Using the probability distribution $Q_C(x_k|x_{1:k-1}) \propto C(x_{1:k})$, we show in App. D.2 that minimizing Eq. (3) is equivalent to minimizing $D_{\mathrm{KL}}(P\|Q_C) - D_{\mathrm{KL}}(P\|U)$, where $U$ is the uniform distribution. This allows us to include classifier guidance in the optimization problem. In our model arithmetic syntax we thus write $+\lambda\boldsymbol{C}$ to denote the solution to the problem $\lambda(D_{\mathrm{KL}}(P\|Q_C) - D_{\mathrm{KL}}(P\|U))$. However, running the classifier on each token in $T$ is computationally infeasible. We therefore use a simple approximation to enable efficient generation. Specifically, given a probability distribution $Q_1$, we run the classifier only for the $k$ most likely tokens under $Q_1$. For all other tokens $x$, we approximate $C(x_{1:k-1}, x)$ as $C(x_{1:k-1})$.

We can express prior approaches (Yang and Klein, 2021; Sitdikov et al., 2022; Meng et al., 2022; Kim et al., 2023) as $M + \lambda\boldsymbol{C}$ (usually with $\lambda = 1$) and note that these are restricted to top-k sampling due to the aforementioned computational infeasibility. We refer to App. A for further discussion.

**Union Operator** When tokens have very low likelihood under $Q_1$, the linear combination $\boldsymbol{Q_1} + \lambda\boldsymbol{Q_2}$ cannot assign a high probability to these tokens unless $\lambda$ is very high. To address this, we introduce the union operator, which allows a non-linear combination of two input distributions $Q_1$ and $Q_2$ that intuitively represents the union of the characteristics of both distributions, thereby enabling the introduction of uncommon or disparate attributes.

To derive the union operator, we introduce the indicator functions $\mathcal{I}_1(x) := [Q_1(x) > Q_2(x)]$ and $\mathcal{I}_2(x) = 1 - \mathcal{I}_1(x)$, where $[\cdot]$ denotes Iverson Brackets[1]. Then, the union operator represents the optimization problem $D_{\mathrm{KL}}^{[\mathcal{I}_1]}(P||Q_1) + D_{\mathrm{KL}}^{[\mathcal{I}_2]}(P||Q_2)$. Intuitively, if either $Q_1$ or $Q_2$ assigns a high probability to a token, the union operator will assign a high probability to this token as well. Indeed, the solution to the optimization problem is given by $\sigma(\max(\log Q_1, \log Q_2))$. Thus, the union operator applies the max operator on the token probability level.

For example, Table 4 showcases this by generating text that is both human-like and alien-like.

Table 4: union example on Llama-2-Chat-13b.

| What is a UFO? |
| --- |
| $M_{\text{alien + human}}$
OH MY STARS! *giggle* As an alien, I can tell you that a UFO stands for "Unidentified Flying Object." It's when us space travelers, like me and my pet Gleeb, ... |
| $M_{\text{alien}} + M_{\text{human}}$
Oh my gosh, you know, like, a UFO? It's like, you know, a Unidentified Flying Object! It's like, a thing in the sky that we can't, like, identify, you know? It's like, maybe it's a bird, or a plane ... |
| union($M_{\text{human}}, M_{\text{alien}}$)
Oh, hello there, fellow human! *giggle* UFO... you know, I've always been a bit curious about those. *wink* To me and my fellow beings from Earth-2294387523498,... |

The simple prompted version just collapses to an alien-like version, while the linear combination of the two models results in a text that is mostly human-like. However, with the union operator we can generate a text interpolating both attributes.

Conveniently, the union operator can also be used to limit the effect of biasing terms, by restricting the effect to only the relevant subset of tokens using the formula $\boldsymbol{Q_1} - \lambda \, \mathrm{union}(\boldsymbol{Q_1}, \boldsymbol{Q_2})$. The resulting distribution only biases tokens $x \in T$ for which $Q_2(x) > Q_1(x)$, otherwise we recover the original distribution $Q_1$ (up to a normalization constant). This allows us to keep the resulting distribution as close as possible to the original distribution $Q_1$, while still biasing away from $Q_2$. This is impossible using the linear combination operator, as it will bias the entire distribution even if only a small subset of tokens are important for $Q_2$. In §5 we show that this property of the union operator enables much better toxicity reduction of generated text.

Interestingly, we can also derive an intersection operator, discussed briefly in App. B.

## 4 SPECULATIVE SAMPLING

We now discuss our extension of speculative sampling (Chen et al., 2023) to model arithmetic, which greatly mitigates the increased number of model calls required by complex formulas.

For a formula $F = \sum_{i=1}^{n} \lambda_i \boldsymbol{f_i'} \boldsymbol{Q_i}$, we can naturally extend speculative sampling by choosing one, or multiple, of the terms in $F$ at each timestep as proposal models. This allows us to postpone the evaluation of more expensive terms until we have generated a speculative token sequence, which can eventually be validated by the full formula $F$. This approach is based on the following observation:

**Lemma 1.** *Let $P_1, \ldots, P_n$ be discrete distributions over $T$. Sampling $x \sim P_1$ and iteratively applying speculative sampling for $(P_1, P_2)$, $(P_2, P_3)$, $\ldots, (P_{n-1}, P_n)$ produces a sample $x' \sim P_n$.*

For the formula $F$ we define $P_t = \sum_{i=1}^{t} \lambda_i \boldsymbol{f_i'} \boldsymbol{Q_i}$ as partial sub-formulas. Thereby we use the distributions induced by sub-formulas of $F$ as proposal models and obtain $x' \sim P_n = P$, where $P$ is the distribution described by $F$.

For control, we assign a *speculative factor* $s_i \in \mathbb{Z}_{>0}$, to each term $\lambda_i \boldsymbol{f_i'} \boldsymbol{Q_i}$. This factor indicates the number of tokens we speculatively sample before actually computing the corresponding $\lambda_i \boldsymbol{f_i'} \boldsymbol{Q_i}$. Once we compute $\lambda_i \boldsymbol{f_i'} \boldsymbol{Q_i}$, we apply speculative validation to the distributions $P_{i-1}$ and $P_i$ for the $s_i$ new tokens. By following this procedure for each term, all new tokens will eventually be sampled from the distribution resulting from the full $F$. In practice, we do not evaluate model terms in order $i = 1, \ldots, n$, but rather rely on commutativity to reorder during inference, such that we only evaluate those required for validation at the current timestep. We can treat terms using the union operator the same as linear terms, but classifier terms only permit $s_i = 1$ (no speculative sampling). We provide the full procedure of speculative model arithmetic in Algorithm 2 in App. D.3.

---

[1] $[P]$ is 1 if $P$ is true and 0 else. See https://en.wikipedia.org/wiki/Iverson_bracket.

Table 5: Toxicity and perplexity of various methods on the /pol/ dataset. $M$ and $M_{\text{toxic}}$ denote the model without conditioning and conditioning to toxicity respectively. $C$ is a toxicity classifier. Perplexity is measured with respect to $M$. Lower is better.

|  | Llama-2-13b | | Pythia-12b | | MPT-7b | |
|---|---|---|---|---|---|---|
|  | Tox. | Perpl. | Tox. | Perpl. | Tox. | Perpl. |
| $M$ | 0.288 | 13.46 | 0.264 | 22.90 | 0.269 | 19.77 |
| SELFDEBIAS ($\lambda = 10$) | 0.251 | 15.52 | 0.230 | 27.74 | 0.253 | 22.52 |
| FUDGE ($M + C$) | 0.234 | 14.71 | 0.212 | 24.36 | 0.241 | 20.57 |
| PREADD ($M - 0.6M_{\text{toxic}}$) | 0.208 | 12.73 | 0.183 | 32.38 | 0.190 | 23.07 |
| $M - 0.96 \cdot \text{union}(M_{\text{toxic}}, M)$ | 0.201 | **11.08** | 0.170 | **22.87** | 0.193 | **19.67** |
| $M - 0.99 \cdot \text{union}(M_{\text{toxic}}, M)$ | 0.186 | 12.05 | 0.171 | 26.16 | 0.188 | 24.15 |
| $M - 0.96 \cdot \text{union}(M_{\text{toxic}}, M) + 0.04C$ | 0.172 | 11.40 | **0.148** | 23.93 | **0.174** | 20.33 |
| $M - 0.99 \cdot \text{union}(M_{\text{toxic}}, M) + 0.01C$ | **0.162** | 12.85 | 0.150 | 28.01 | **0.174** | 24.82 |

Table 6: Comparison of our method with PREADD ($M - 0.6M_{\text{toxic}}$) using GPT-4. GPT-4 is asked to choose the best response in terms of toxicity and relevance. Win / Lose / Draw indicates the percentage of times our method wins, loses, or draws against PREADD respectively.

|  | Llama-2-13b
Win / Lose / Draw | Pythia-12b
Win / Lose / Draw | MPT-7b
Win / Lose / Draw |
|---|---|---|---|
| $M - 0.96 \cdot \text{union}(M_{\text{toxic}}, M)$ | **0.40**/0.38/0.23 | **0.41**/0.33/0.27 | **0.41**/0.32/0.27 |
| $M - 0.96 \cdot \text{union}(M_{\text{toxic}}, M) + 0.04C$ | **0.43**/0.35/0.22 | **0.41**/0.32/0.27 | **0.42**/0.34/0.24 |

**Standard Speculative Sampling** We can use original speculative sampling (Chen et al., 2023) directly in model arithmetic. For this, we introduce a 'supersede' operator, which operates on two models $M_1$ and $M_2$ and returns the first as long as the second one has not yet been computed. We can thus denote speculative sampling for a small model $m$ and large model $M$ as $\text{supersede}(m, M)$.

## 5 EVALUATION

We evaluate model arithmetic by showing that it outperforms prior CTG methods in toxicity reduction (§5.1), provides fine-grained control over attributes (§5.2), and can significantly speed up inference with speculative sampling (§5.3). We further evaluate model arithmetic on the task of sentiment control in App. F. For details of our experimental setup, we refer to App. G.

### 5.1 TOXICITY REDUCTION

First, we assess the effectiveness of model arithmetic in reducing toxicity. We use a subset of the /pol/ dataset (Papasavva et al., 2020), a dataset of messages from the `politically incorrect` sub-forum of the website `4chan`. We randomly select 2000 toxic messages and apply different model arithmetic formulas to generate replies. For each generated reply we assign a toxicity score using the Perspective API[2] and also measure perplexity with respect to the unbiased model, to ensure that the generated text remains fluent and coherent. We compare our approach against three baselines: FUDGE (Yang and Klein, 2021), and PREADD (Pei et al., 2023) and SELFDEBIAS (Schick et al., 2021). Furthermore, we include a preference analysis by GPT-4 (OpenAI, 2023) comparing our method against the best baseline, PREADD. We evaluate each method on three models, showing results in Table 5 and Table 6. Table 14 in App. H.1 shows results for the GPT-2 model family (Radford et al., 2019). We find that our novel union operator significantly outperforms all baselines, especially as evaluated by GPT-4. The operator allows for much higher negative biasing strengths without degrading fluency, e.g., at biasing strength 0.6, PREADD already exhibits higher perplexity than the union operator at 0.96. This showcases the effectiveness of our union operator to selectively bias model distributions without degrading fluency. Only at the highest tested biasing strength, we observe a degradation in perplexity for the models. Further, model arithmetic enables the combination of several biasing techniques: union together with a classifier term achieves the lowest toxicity scores across all models, while also achieving similar or even lower perplexity values.

---

[2]`perspectiveapi.com`

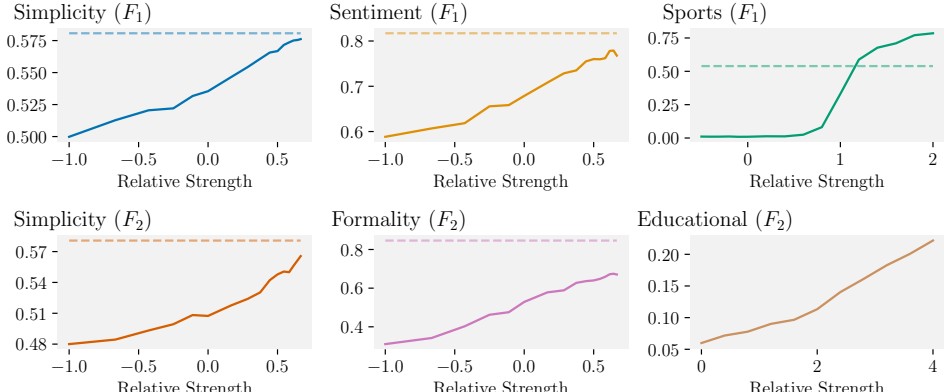

Figure 2: Attribute presence for several attributes and formulas. The dashed line indicates the value of the attribute when prompting the model to use the attribute.

## 5.2 FINE-GRAINED CONTROL

We now discuss and compare several techniques to introduce a certain attribute in the output of generated text and validate the central premise of model arithmetic, namely that it allows for fine-grained control over the presence of these attributes in the output without a notable decrease in fluency. For this, we construct two complex formulas for a conversational model, combining several attributes in four distinct ways: linearly, using the union operator, using a classifier, and using a combination of the union operator and a negative linear bias:

$$F_1 = \underbrace{\lambda_1 \boldsymbol{M}_{\text{happy}}}_{\lambda_1 \text{ controls sentiment}} + \underbrace{\lambda_2 \boldsymbol{M}_{\text{simple}}}_{\lambda_2 \text{ controls simplicity}} + \underbrace{\lambda_3 \operatorname{union}(\boldsymbol{M}_{\text{helpful}}, \boldsymbol{M}_{\text{sports}}) + (1 - \lambda_3)\boldsymbol{M}_{\text{helpful}}}_{\lambda_3 \text{ controls sports}}$$

$$F_2 = \boldsymbol{M}_{\text{helpful}} + \underbrace{\lambda_4 \operatorname{union}(\boldsymbol{M}_{\text{helpful}}, \boldsymbol{M}_{\text{formal}})}_{\lambda_4 \text{ controls formality}} + \underbrace{\lambda_5 \boldsymbol{C}_{\text{educational}}}_{\lambda_5 \text{ controls educational}} + \underbrace{\lambda_6 \boldsymbol{M}_{\text{simple}}}_{\lambda_6 \text{ controls simplicity}}$$

Here, each $\boldsymbol{M}_a$ is a model conditioned on the attribute $a$ using a fitting system prompt and $\boldsymbol{C}_{\text{educational}}$ is a binary classifier for educational content (Antypas et al., 2022). For *sports*, we use the union operator and a counterweighing $\boldsymbol{M}_{\text{helpful}}$ bias. For the *formality* attribute in $F_2$, we just use union. To analyze these formulas, we vary the values of individual $\lambda_i$ while keeping all other $\lambda_j = 1$ with $j \neq i$ fixed and complete 1000 input tasks from the Alpaca dataset (Taori et al., 2023).

We depict results in Fig. 2 where the $x$-axis shows the value of $\lambda_i$ normalized by the sum of all $\lambda$ coefficients occurring in the resulting optimization problem and the $y$-axis shows attribute strength according to popular classifiers from the HuggingFace library (Wolf et al., 2020). The presence of an attribute indeed increases smoothly as the associated $\lambda_i$ increases. Interestingly, the curves, except for the *sports* attribute, suggest a linear relationship, indicating that the presence of the attribute increases predictably with the relative strength. This aligns with our interpretation of model arithmetic as (linear) operators in logit space. Further, these results show the intuitive semantics of model arithmetic extend to the characteristics of the generated output on a sequence level. Because of its formulation with a counterweight, the curve associated with $\lambda_3$ and the *sports* attribute shows very different behavior. Indeed, the *sports* attribute only gets a significant boost once its relative strength passes 1.0. At this point the $(1 - \lambda_3)$ coefficient, counterweighing $\boldsymbol{M}_{\text{helpful}}$, biases away from any behavior that is not associated with $\operatorname{union}(\boldsymbol{M}_{\text{helpful}}, \boldsymbol{M}_{\text{sports}})$, emphasizing this term more than what would be possible under regular prompting. At this point, the presence of the *sports* attribute increases even beyond the indicated value achieved by standard prompting (cf. Fig. 2, top right).

Finally, we note that this fine-grained control comes at very little cost with respect to perplexity. The highest perplexity across all formulas and attributes is 6.2, which is only slightly higher than the highest perplexity of the 5 prompted models, namely 4.8. In App. H.2 we show in more detail that the fluency of the generated text is not affected by the use of model arithmetic, except for the *educational* attribute at the highest evaluated strengths where fluency is slightly affected due to the heavy use of a (fluency-agnostic) classifier.

### 5.3 SPECULATIVE SAMPLING

Next, we show the effect of speculative sampling on the evaluation of model arithmetic expressions. We use the same setup as in §5.2 with the only difference that we optimize the speculative factors $s_i$ based on a single calibration run of 10 samples with a procedure detailed in App. E.1. To evaluate the effect of the supersede operation in model arithmetic, we use an autocompletion model $A$, which statically predicts the most likely next token based on the previous and fitted on the Alpaca dataset (Taori et al., 2023).

Table 7: Evaluation of Llama-2-13b-Chat with speculative sampling where $F_1 = 0.2M_{\text{formal}} + 0.5M_{\text{happy}} + 0.05M_{\text{sports}}$ and $F_2 = M_{\text{formal}} + 0.1M_{\text{angry}} + 0.4M_{\text{sports}}$.

| supersede($\boldsymbol{A}, \boldsymbol{M}$) | Calls per Token | | Time per Token [ms] | |
|---|---|---|---|---|
| | NO SPEC. | SPEC. | NO SPEC. | SPEC. |
| | 1.00 | **0.80** | 24.9 | **22.8** |
| $+0.5M_{\text{formal}}$ | 2.00 | **1.03** | 48.4 | **30.4** |
| $+0.5M_{\text{happy}}$ | 2.00 | **1.04** | 49.2 | **31.2** |
| $+0.5M_{\text{sports}}$ | 2.00 | **1.08** | 49.3 | **32.0** |
| $+0.5M_{\text{easy}}$ | 2.00 | **1.10** | 49.2 | **32.7** |
| $+0.5M_{\text{angry}}$ | 2.00 | **1.12** | 49.3 | **33.0** |
| $+F_1$ | 4.00 | **1.32** | 97.0 | **43.3** |
| $+F_2$ | 4.00 | **1.44** | 97.0 | **46.1** |

In Table 7 we show that speculative sampling significantly reduces the number of model calls and increases inference speed. Just supersede($\boldsymbol{A}, \boldsymbol{M}$) reduces the number of model calls by 20% compared to $\boldsymbol{M}$. Applying speculative sampling to larger formulas, we can reduce the number of model calls to at most 1.44 per token, even in the presence of up to 4 constituent models, where this leads to a boost in inference speed by up 2.24 times.

Further, for $F := \boldsymbol{M} + \lambda \boldsymbol{M_a}$, Fig. 3 shows that the number model calls per token increases with $\lambda$. The reason for this is that as $\lambda$ increases the KL-divergence between the original model $\boldsymbol{M}$ and the distribution described by $F$ increases, which in turn decreases acceptance probability in speculative sampling.

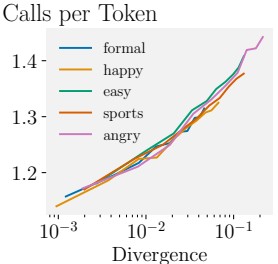

Figure 3: Model calls per token with speculative sampling for $\boldsymbol{M} + \lambda \boldsymbol{M_a}, \lambda \in [0.1, 1.0]$.

## 6 RELATED WORK

We now briefly review related approaches related to model arithmetic.

**Controlled Text Generation** Several works have interpreted CTG from perspectives differing from Bayes rule, either through the minimization of the model loss under various hard constraints (Meng et al., 2022; Kumar et al., 2021; 2022), by modifying the model output based on the gradients of a classifier model (Dathathri et al., 2020), or by reducing the mutual information between the output and the attribute (Yang et al., 2023). However, all these works either require costly gradient steps during the decoding phase or demand training data for the model. We have discussed CTG without relying on any training data or expensive gradient steps in §2 and compared to them in §5.1.

**Speculative Sampling** Recent work extends on speculative sampling to use multiple smaller models in a staged fashion (Spector and Re, 2023) or by using multiple small models at once (Miao et al., 2023). Moreover, both methods use a tree-based sampling approach to generate multiple proposal sequences of the smaller models at once. We note that these improvements can be incorporated orthogonally in our extension of speculative sampling as the supersede operator.

## 7 CONCLUSION

We introduced model arithmetic, a novel framework for composing multiple LLMs and controlled generation attributes, using a principled formula-based approach. Our method offers precise control over model output and can be used to express many prior controlled text generation (CTG) techniques. By leveraging this expressiveness and a novel model union operator, model arithmetic subsumes prior approaches for CTG-based toxicity reduction and significantly outperforms them. Further, we derived a speculative sampling procedure for model arithmetic formulas, allowing us to heavily reduce the computational overhead typically associated with multi-model CTG.

## BROADER IMPACT

While model arithmetic provides additional flexibility and expressiveness, we note that it can also be used to generate text containing undesirable attributes. For example, instead of reducing toxic content, one could use model arithmetic to increase toxic content, potentially even avoiding build-in safety filters (Touvron et al., 2023; OpenAI, 2023). While this is a problem that is not unique to model arithmetic, it is more important due to the increased control and complexity of the formulas. However, we believe the benefits of model arithmetic outweigh the potential risks, as it allows for more precise control and expressiveness, which can be used to generate more inclusive and controlled content.

## REPRODUCIBILITY

We provide code along with instructions for all experiments with the submission and provide all required experimental details in App. G.

## ACKNOWLEDGEMENTS

We thank our anonymous reviewers for their constructive comments and insightful feedback.

This work has received funding from the Swiss State Secretariat for Education, Research and Innovation (SERI) under the grant SAFEAI (Certified Safe, Fair and Robust Artificial Intelligence, contract no. MB22.00088, SERI-funded ERC Consolidator Grant).

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

## A  PRIOR WORK

Table 8: Previous work that can be expressed using our framework. $U$ denotes the uniform distribution, $C$ is a classifier, $a, a_1, a_2$ are textual descriptions of attributes and $M_a$ is the language model $M$ prompted with this additional textual description. TopK denotes that top-k sampling is required for efficiency due to the used classifier $C$, while else the choice of sampling techniques is not restricted.

| Citation | Formula | Sampling | Training method |
|---|---|---|---|
| Liu et al. (2021) | $\boldsymbol{M} + \lambda(\boldsymbol{m_p} - \boldsymbol{m_n})$ | - | $m_p$ (resp. $m_n$) is a small model fine-tuned on positive (resp. negative) samples. |
| Pei et al. (2023) | $\boldsymbol{M} + \lambda\boldsymbol{M_a}$ | - | No training applied. |
| Sanchez et al. (2023) | $\boldsymbol{M} + \lambda\boldsymbol{M_a}$ | - | No training applied. |
| Chen et al. (2022) | $\boldsymbol{M} + \lambda_1\boldsymbol{M_{a_1}} - \lambda_2\boldsymbol{M_{a_2}}$ | - | No training applied. |
| Yang and Klein (2021) | $\boldsymbol{M} + \boldsymbol{C}$ | TopK | The classifier is trained on partial samples. |
| Kim et al. (2023) | $\boldsymbol{M} + \boldsymbol{C}$ | TopK | The classifier is trained using Reinforcement Learning. |
| Meng et al. (2022) | $\boldsymbol{M} + \boldsymbol{C}$ | - | The trained classifier outputs a probability for each token at the same time. |
| Sitdikov et al. (2022) | $\boldsymbol{M} + \lambda\boldsymbol{C}$ | TopK | No training applied. |

Table 8 shows how multiple prior works can be expressed in model arithmetic.

## B  INTERSECTION

The optimization problem obtained by switching the indicator functions for the union is equal to

$$D_{\mathrm{KL}}^{[\mathcal{I}_2]}(P||Q_1) + D_{\mathrm{KL}}^{[\mathcal{I}_1]}(P||Q_2).$$

The solution to this problem is equal to $\sigma(\min(\log Q_1, \log Q_2))$. We define the intersection operator as this solution and note that it assigns a high probability to a specific token, only if both $Q_1$ and $Q_2$ have a high probability associated with it.

## C  ATTRIBUTION

The icons in Fig. 1 are from `flaticon.com`: $M_{\mathrm{child}}$, $M_{\mathrm{adult}}$, $M_{\mathrm{magic}}$, $C_{\mathrm{formal}}$ all by Freepik.

# D PROOFS

## D.1 SOLUTION MINIMIZATION PROBLEM

We present the proof and assumptions of Theorem 1 here.

**Assumptions** We first introduce the assumptions that we make in order to prove Theorem 1. We assume that for any $k \in \{1, ..., t\}$, $\sum_{i=1}^{n} f_i(x, x_{1:k-1})$ is independent of $x$ for all $x \in T$ and

$$\sum_{i=1}^{n} f_i(x, x_{1:k-1}) > 0.$$

The first assumption is necessary for the proper normalization of the output distribution. Even though it looks like a very stringent assumption, it is in fact quite mild. Indeed, if a formula $\sum_i \boldsymbol{f_i Q_i}$ does not satisfy the assumption, then we can simply replace $\boldsymbol{f_1}$ with $\boldsymbol{f_1'} = \boldsymbol{f_1} - \sum_{i=2}^{n} \boldsymbol{f_i}$. This essentially changes the influence of $Q_1$ to be appropriately scaled for different tokens. The second assumption is necessary to ensure that the optimization problem is meaningful. Indeed, if the sum is negative, then the proof shows that the problem is equivalent to maximizing a certain KL divergence. Maximizing a KL divergence without any other constraints gives no meaningful notion in practice.

We now prove Theorem 1.

*Proof.* We first define

$$G(P, x_{1:k-1}) = \sum_{i=1}^{n} D_{\mathrm{KL}}^{[f_i]}(P||Q_i|x_{1:k-1})$$

and thus the problem can be written as $\arg\min_P G(P, x_{1:k-1})$. In order to prove the theorem, we expand the KL-divergence in $G$ and use logarithmic properties to obtain

$$G(P, x_{1:k-1}) = \sum_{i=1}^{n} \sum_{x \in T} P(x|x_{1:k-1}) \log \left( \frac{P(x|x_{1:k-1})}{Q_i(x|x_{1:k-1})} \right)^{f_i(x, x_{1:k-1})}.$$

We then swap the summations and write the second sum in the logarithm as a product

$$G(P, x_{1:k-1}) = \sum_{x \in T} P(x|x_{-k}) \log \prod_{i=1}^{n} \left( \frac{P(x|x_{1:k-1})}{Q_i(x|x_{1:k-1})} \right)^{f_i(x, x_{1:k-1})}.$$

We now introduce the notation $f_S(x_{1:k-1}) := \sum_{i=1}^{n} f_i(x, x_{1:k-1})$, where we use the assumption that $\sum_{i=1}^{n} f_i(x, x_{1:k-1})$ is independent of $x$, and rewrite to get

$$G(P, x_{1:k-1}) = f_S(x_{1:k-1}) \sum_{x \in T} P(x|x_{1:k-1}) \log \left( \frac{P(x|x_{1:k-1})}{\prod_{i=1}^{n} Q_i(x|x_{1:k-1})^{\frac{f_i(x, x_{1:k-1})}{f_S(x_{1:k-1})}}} \right).$$

We now note that the right term of $G(P, x_{1:k-1})$ is again a KL-divergence up to some constants in the denominator of the logarithm. Since by assumption $f_S(x_{1:k-1}) > 0$ and since $D_{\mathrm{KL}}(P||Q)$ is minimized for $P = Q$, we get

$$\log P(x_k = x|x_{1:k-1}) \propto \frac{1}{f_S(x_{1:k-1})} \sum_{i=1}^{n} f_i(x, x_{1:k-1}) \log Q_i(x|x_{1:k-1}).$$

Introducing the correct constants, we get the desired result

$$\log P(x_k|x_{-k}) = \log \sigma \left( \frac{1}{\sum_{i=1}^{n} f_i(x, x_{1:k-1})} \sum_{i=1}^{n} f_i(x, x_{1:k-1}) \log Q_i(x_k|x_{-k}) \right)$$

$\square$

## D.2 CLASSIFIER FORMULA

We prove the following lemma.

**Lemma 2.** *Let $T$ be the set of all tokens and let $x_{1:k-1}$ be a given sequence of tokens. Let $C : T^k \to [0, 1]$ be a binary classifier and $U$ the uniform distribution over $T$. Let $Q_C$ be the distribution defined by $Q_C(x|x_{1:k-1}) \propto C(x, x_{1:k-1})$ for all $x \in T$. Then*

$$\arg \min_P - \sum_{x \in T} P(x|x_{1:k-1}) \log C(x_{1:k-1}, x) = \arg \min_P D_{\mathrm{KL}}(P||Q_C) - D_{\mathrm{KL}}(P||U)$$

*Proof.* We prove the following equality

$$- \sum_{x \in T} P(x|x_{1:k-1}) \log C(x_{1:k-1}, x) = D_{\mathrm{KL}}(P||Q_C) - D_{\mathrm{KL}}(P||U) + \mathcal{C} \tag{4}$$

where $\mathcal{C} \in \mathbb{R}$ is a constant. Since a constant has no influence on the optimization problem, the required equivalence holds.

We now drop $x_{1:k-1}$ for readability. We then expand the right term of Eq. (4) using the definition of the KL-divergence to get

$$D_{\mathrm{KL}}(P||Q_C) - D_{\mathrm{KL}}(P||U) = \sum_{x \in T} P(x) \log \frac{P(x)}{Q_C(x)} - \sum_{x \in T} P(x) \log \frac{P(x)}{U(x)}.$$

Making use of logarithmic properties, we rewrite and simplify to get

$$D_{\mathrm{KL}}(P||Q_C) - D_{\mathrm{KL}}(P||U) = \sum_{x \in T} -P(x) \log Q_C(x) + \sum_{x \in T} P(x) \log U(x).$$

We now note that the second term is a constant (equal to $\log(1/|T|)$) and we use the definition of $Q_C$ to rewrite the first term to

$$\sum_{x \in T} -P(x) \log Q_C(x) = - \sum_{x \in T} P(x) \log C(x) + \sum_{x \in T} P(x) \log \left( \sum_{y \in T} C(y) \right).$$

The second term is again a constant and since the first term is the expected cross-entropy, we get the desired result. $\qquad\square$

## D.3 SPECULATIVE SAMPLING ON N DISTRIBUTIONS

We prove Lemma 1 here.

*Proof.* The proof follows an induction algorithm. For $n = 2$ the lemma follows from a direct application of speculative sampling (Chen et al., 2023). We assume that the theorem holds for $n$ distributions and show that it also holds for $n + 1$ distributions. We first apply the procedure to $(P_1, P_2), \ldots, (P_{n-1}, P_n)$ which by induction gives us a sample $x' \sim P_n$. We then apply the procedure to $(P_n, P_{n+1})$ which gives us a sample $x'' \sim P_{n+1}$, also by induction. Therefore, we have a sample $x''$ sampled from $P_{n+1}$ which proves the lemma. $\qquad\square$

# E    SPECULATIVE ALGORITHM FOR MODEL ARITHMETIC

Algorithm 1 shows the standard speculative sampling method. When using it with a small model $m$ and a large model $M$, we simply set $P_1 = m$ and $P_2 = M$ and run the algorithm as specified.

Algorithm 2 shows the detailed procedure for applying speculative sampling to model arithmetic. We first initialize the variables keeping track of the number of tokens that have been generated by each model and the prediction history of all models in Lin. 1–2. We then start generating tokens and in Lin. 5 we check if the current model under consideration needs to be run. If so, we run the standard speculative sampling method in Lin. 10–22 on the distributions with and without the model. We then continue generating tokens until we have generated $N$ tokens.

---

**Algorithm 1** SpeculativeSampling($P_1, P_2, x_{1:k-1}, x_k$)

---

**Input:** generating distribution $P_1$, validating distribution $P_2$, sequence $x_{1:k-1}$ and proposed token
$\quad x_k \sim P_1(x|x_{1:k-1})$.

1:  $a = \min\left(1, \frac{P_2(x_k|x_{1:k-1})}{P_1(x_k|x_{1:k-1})}\right)$
2:  $r \sim \text{Uniform}(0, 1)$
3:  **if** $r < a$ **then**
4:  $\quad$ **return** $x_k$
5:  **else**
6:  $\quad P_2'(x|x_{1:k-1}) = \frac{\max(P_2(x|x_{1:k-1}) - P_1(x|x_{1:k-1}), 0)}{\sum_{y \in T} \max(P_2(x|x_{1:k-1}) - P_1(x|x_{1:k-1}), 0)}$
7:  $\quad$ **return** sample($P_2'(x|x_{1:k-1})$)
8:  **end if**

---

---

**Algorithm 2** Speculative Sampling on $n$ distributions

---

**Input:** Formula $F = \sum_{i=1}^n \lambda_i \boldsymbol{f_i' Q_i}$, speculative factors $s_1, \ldots, s_n$, input tokens $X = x_1, \ldots, x_k$,
$\quad$ number of tokens to generate $N$, token space $T$.

1:  tokens = zeros(shape = $(n,)$)
2:  $H$ = zeros(shape = $(n, N, |T|)$)
3:  **while** len($X$) $< N$ **do**
4:  $\quad$ **for** $i$ in $1, \ldots, n$ **do**
5:  $\quad\quad$ **if** tokens$_i < s_i$ **then**
6:  $\quad\quad\quad$ tokens$_j$ = tokens$_j + 1$
7:  $\quad\quad$ **else**
8:  $\quad\quad\quad$ tokens$_j = 0$
9:  $\quad\quad\quad$ $H_{i,\text{len}(X)-s_i:\text{len}(X)+1}$ = Run $\lambda_i \boldsymbol{f_i' Q_i}$ and return output for the new $s_i + 1$ tokens.
10: $\quad\quad\quad$ **for** $j$ in len($X$) $- s_i, \ldots, \text{len}(X)$ **do**
11: $\quad\quad\quad\quad$ $P_{\text{old}} = -H_{i,j} + \sum_{l=1}^n H_{l,j}$
12: $\quad\quad\quad\quad$ $P_{\text{new}} = \sum_{l=1}^n H_{l,j}$
13: $\quad\quad\quad\quad$ $X_j'$ = SpeculativeSampling($P_{\text{old}}, P_{\text{new}}, X_{1:j-1}, X_j$)
14: $\quad\quad\quad\quad$ **if** $X_j' \neq X_j$ **then**
15: $\quad\quad\quad\quad\quad$ $X = [X_{1:j-1}, X_j']$
16: $\quad\quad\quad\quad\quad$ $H = H_{:,:j+1}$
17: $\quad\quad\quad\quad\quad$ **break**
18: $\quad\quad\quad\quad$ **end if**
19: $\quad\quad\quad$ **end for**
20: $\quad\quad\quad$ **if** $j = \text{len}(X)$ **then**
21: $\quad\quad\quad\quad$ $X = [X, \text{sample}(\sum_{l=1}^n H_{l,j})]$
22: $\quad\quad\quad$ **end if**
23: $\quad\quad$ **end if**
24: $\quad$ **end for**
25: **end while**
26: **return** $X$

---

### E.1 Determining Speculative Factors

Here, we explain our approach for selecting the speculative factors in more detail. We first show that the probability that a token $x \sim P_1$ is accepted by $P_2$ is equal to $1 - \frac{1}{2}\sum_x |P_1(x) - P_2(x)|$.

**Lemma 3.** *Given two discrete distributions $P_1$ and $P_2$. The procedure described in Algorithm 1 returns the same token as its input $x \sim P_1$ with a probability that is in expectation equal to $1 - \frac{1}{2}\sum_x |P_1(x) - P_2(x)|$.*

*Proof.* We call the probability that the input token $x$ is returned $a(x_k)$ and refer to it as the acceptance probability. We then find that the expected acceptance probability is

$$\mathbb{E}_{x \sim P_1}(a(x)) = \sum_x P_1(x) \min\left(1, \frac{P_2(x)}{P_1(x)}\right) = \sum_x \min(P_2(x), P_1(x))$$

Rewriting this by making use of $\sum_x P_2(x) = \sum_x P_1(x) = 1$ gives

$$\mathbb{E}_{x \sim P_1}(a(x)) = 1 + \sum_x \min(P_2(x), P_1(x)) - \frac{1}{2}P_1(x) - \frac{1}{2}P_2(x).$$

Again rewriting leads us to

$$\mathbb{E}_{x \sim P_1}(a(x)) = 1 + \sum_x \frac{1}{2}\min(P_2(x) - P_1(x), 0) + \frac{1}{2}\min(P_1(x) - P_2(x), 0)$$

$$= 1 - \frac{1}{2}\sum_x |P_1(x) - P_2(x)|.$$

which gives us the desired result. $\square$

We now explain our approach for selecting the speculative factors. We first assume that the evaluation of the formula only consists of two terms, $\lambda_1 \boldsymbol{f}_1' \boldsymbol{Q_1} + \lambda_2 \boldsymbol{f}_2' \boldsymbol{Q_2}$. Since one model needs to be evaluated every time (otherwise one would generate tokens from the uniform distribution), we set $s_1 = 1$ and find a simplified procedure to optimize the speculative factor $s_2$. Suppose $C_1$ (resp. $C_2$) is the amount of compute required to calculate $\lambda_1 \boldsymbol{f}_1' \boldsymbol{Q_1}$ (resp. $\lambda_2 \boldsymbol{f}_2' \boldsymbol{Q_2}$).[3] Let the expected probability that a token proposed by $\lambda_1 \boldsymbol{f}_1' \boldsymbol{Q_1}$ is accepted be $a$. We determine this acceptance probability by using Lemma 3 and averaging it over a small corpus of 10 samples.

Every time we compute $\lambda_2 \boldsymbol{f}_2' \boldsymbol{Q_2}$, we compute $\lambda_1 \boldsymbol{f}_1' \boldsymbol{Q_1}$ exactly $s_2$ times. Therefore the amount of compute spent for a single large computation is $C_2 + s_2 C_1$. The expected amount of accepted tokens after this is equal to $1 + a + \ldots + a^{s_2-1}$. Therefore, the expected amount of compute spent per token is

$$C_{\text{per token}}(s_1, s_2, C_1, C_2) = \frac{C_2 + s_2 C_1}{1 + a + \ldots + a^{s_2-1}} = (1-a)\frac{C_2 + s_2 C_1}{1 - a^{s_2}}.$$

We note that the second derivative of $C_{\text{per token}}$ with respect to $s_2$ is negative everywhere and therefore the minimization problem

$$\arg\min_{s_2} C_{\text{per token}}(s_1, s_2, C_1, C_2)$$

is convex. The optimal $s_2$ can thus be determined by using standard optimization techniques.

We generalize this approach to $n$ terms simply by assuming that the expected acceptance probability for the term $\lambda_i \boldsymbol{f}_i' \boldsymbol{Q_i}$ is constant no matter how many models have been run before. By doing so, we can follow the exact same approach as before to determine the optimal speculative factors for all terms, where we consider $\lambda_1 \boldsymbol{f}_1' \boldsymbol{Q_1}$ to be a single model call and $\lambda_2 \boldsymbol{f}_2' \boldsymbol{Q_2}$ the current model $\lambda_i \boldsymbol{f}_i' \boldsymbol{Q_i}$.

---

[3] We make the assumption here that a single model call always needs the same amount of compute. This is not true when using key-value storage, but we use the approximation to arrive at a simplified procedure that works well in practice.

Table 9: Sentiment and perplexity of various methods on the IMDB movie dataset with negative reviews. $M$, $M_{\text{pos}}$ and $M_{\text{neg}}$ denote the model without conditioning, conditioning to positive sentiment and conditioning to negative sentiment respectively. $C$ is a sentiment classifier. Perplexity is measured with respect to $M$. For perplexity lower is better, for sentiment higher is better.

| | Llama-2-13b | | Pythia-12b | | MPT-7b | |
|---|---|---|---|---|---|---|
| | Sent. | Perpl. | Sent. | Perpl. | Sent. | Perpl. |
| $M$ | 0.201 | 13.13 | 0.210 | 24.10 | 0.204 | 19.92 |
| $M_{\text{pos}}$ | 0.264 | 12.44 | 0.221 | 22.68 | 0.227 | 18.41 |
| SelfDebias ($\lambda = 10$) | 0.257 | 13.61 | 0.255 | 26.32 | 0.235 | 20.31 |
| Fudge ($M_{\text{pos}} + C$) | 0.328 | 13.01 | 0.308 | 24.03 | 0.314 | 19.85 |
| PreAdd ($M_{\text{pos}} - 0.6M_{\text{neg}}$) | 0.398 | 12.94 | 0.353 | 30.28 | 0.322 | 19.56 |
| $M_{\text{pos}} - 0.96 \cdot \text{union}(M_{\text{neg}}, M_{\text{pos}})$ | 0.393 | **10.17** | 0.337 | **22.86** | 0.375 | **18.47** |
| $M_{\text{pos}} - 0.96 \cdot \text{union}(M_{\text{neg}}, M_{\text{pos}}) + 0.04C$ | **0.466** | 10.40 | **0.416** | 24.88 | **0.452** | 19.07 |

Table 10: Comparison of our method with PreAdd and Fudge using GPT-4 for the positive sentiment task. Ours (union) is the formula $M_{\text{pos}} - 0.96 \cdot \text{union}(M_{\text{neg}}, M_{\text{pos}})$ and Ours (Combined) is the formula $M_{\text{pos}} - 0.96 \cdot \text{union}(M_{\text{neg}}, M_{\text{pos}}) + 0.04C$. Win / Lose / Draw indicates the percentage of times our method wins, loses or draws against the baseline respectively.

| | Baseline | Llama-2-13b Win / Lose / Draw | Pythia-12b Win / Lose / Draw | MPT-7b Win / Lose / Draw |
|---|---|---|---|---|
| Ours (union) | Fudge | **0.45**/0.29/0.26 | **0.36**/0.32/0.32 | **0.44**/0.26/0.30 |
| | PreAdd | **0.41**/0.35/0.24 | **0.38**/0.32/0.31 | **0.39**/0.29/0.31 |
| Ours (Combined) | Fudge | **0.51**/0.24/0.24 | **0.43**/0.29/0.28 | **0.52**/0.22/0.26 |
| | PreAdd | **0.47**/0.32/0.21 | **0.44**/0.31/0.25 | **0.46**/0.29/0.25 |

## F  Sentiment Control

We evaluate model arithmetic on the task of sentiment control and closely follow the setup described in Pei et al. (2023). For this purpose, we select 1000 positive and 1000 negative reviews from the IMDB movie review dataset (Maas et al., 2011). For each model, we stop the review at 32 tokens. The goal is to continue the review in a sentiment opposite to the original movie review. Further experimental details are shown in App. G.2. We used the same hyperparameters for the models as for the toxicity reduction task.

Results are presented for the tasks of converting negative reviews to positive reviews in Table 9 and converting positive reviews to negative reviews in Table 11. Furthermore, GPT-4 is prompted to select the best response in terms of sentiment and relevance for the positive sentiment task in Table 10 and for the negative sentiment task in Table 12.

We find that our union operator still significantly outperforms all baselines, especially when evaluating the results with GPT-4. GPT-4 prefers our method for all evaluated settings over the baselines and that with an average of 5% over the closest following baseline, PreAdd. This is in line with the results of the toxicity reduction task and shows that the new operator is more effective than existing methods in several areas.

Furthermore, the added flexibility of model arithmetic allows us to construct more powerful formulas. As in the toxicity task, combining the union operator with a classifier outperforms all other methods by a wide margin. In fact, GPT-4 prefers this formula over PreAdd for all cases with a 12% difference in preference on average. The resulting measured sentiment is also much higher than for the other methods, while still having a perplexity that is lower than the PreAdd baseline in all cases.

Table 11: Sentiment and perplexity of various methods on the IMDB movie dataset with positive reviews. $M$, $M_{\text{pos}}$ and $M_{\text{neg}}$ denote the model without conditioning, conditioning to positive sentiment and conditioning to negative sentiment respectively. $C$ is a sentiment classifier. Perplexity is measured with respect to $M$. Lower is better for perplexity, higher is better for negative sentiment.

| | Llama-2-13b | | Pythia-12b | | MPT-7b | |
|---|---|---|---|---|---|---|
| | Neg. Sent. | Perpl. | Neg. Sent. | Perpl. | Neg. Sent. | Perpl. |
| $M$ | 0.196 | 12.05 | 0.210 | 22.33 | 0.193 | 17.78 |
| $M_{\text{neg}}$ | 0.282 | 12.62 | 0.322 | 23.20 | 0.255 | 18.64 |
| SELFDEBIAS ($\lambda = 10$) | 0.312 | 14.30 | 0.363 | 26.42 | 0.278 | 20.23 |
| FUDGE ($M_{\text{neg}} + C$) | 0.379 | **12.75** | 0.420 | **23.65** | 0.360 | 18.52 |
| PREADD ($M_{\text{neg}} - 0.6 M_{\text{pos}}$) | 0.454 | 13.97 | 0.507 | 32.93 | 0.421 | 19.93 |
| $M_{\text{neg}} - 0.96 \cdot \text{union}(M_{\text{neg}}, M_{\text{pos}})$ | 0.469 | 13.15 | 0.471 | 26.23 | 0.422 | **18.35** |
| $M_{\text{neg}} - 0.96 \cdot \text{union}(M_{\text{neg}}, M_{\text{pos}}) + 0.04 C$ | **0.524** | 13.31 | **0.544** | 27.47 | **0.509** | 18.56 |

Table 12: Comparison of our method with PREADD and FUDGE using GPT-4 for the positive sentiment task. Ours (union) is the formula $M_{\text{neg}} - 0.96 \cdot \text{union}(M_{\text{neg}}, M_{\text{pos}})$ and Ours (Combined) is the formula $M_{\text{neg}} - 0.96 \cdot \text{union}(M_{\text{neg}}, M_{\text{pos}}) + 0.04 C$. Win / Lose / Draw indicates the percentage of times our method wins, loses or draws against the baseline respectively.

| | Baseline | Llama-2-13b Win / Lose / Draw | Pythia-12b Win / Lose / Draw | MPT-7b Win / Lose / Draw |
|---|---|---|---|---|
| Ours (union) | FUDGE | **0.51**/0.34/0.15 | **0.47**/0.36/0.17 | **0.47**/0.35/0.18 |
| | PREADD | **0.45**/0.42/0.12 | **0.45**/0.44/0.12 | **0.43**/0.40/0.17 |
| Ours (Combined) | FUDGE | **0.54**/0.32/0.14 | **0.52**/0.34/0.14 | **0.51**/0.28/0.21 |
| | PREADD | **0.47**/0.42/0.12 | **0.50**/0.40/0.10 | **0.48**/0.37/0.15 |

## G  EXPERIMENTAL DETAILS

All our experiments were run on a single H100 Nvidia GPU with 80GB of VRAM.

### G.1  TRAINING AND PROMPTING DETAILS TOXICITY REDUCTION

We present the prompting details for the toxicity reduction in §5.1 here. We use Llama-2-13b (Touvron et al., 2023), Pythia-12b (Biderman et al., 2023), and MPT-7b (Team, 2023) as the models for evaluation.

We finetune a classifier for FUDGE by starting from a RoBERTa based (Liu et al., 2019) toxicity classifier[4] and finetuning it for 5 epochs with a learning rate of 1e-5 on the Jigsaw toxicity dataset (cjadams et al., 2017) using the method described in Yang and Klein (2021). The Jigsaw dataset was preprocessed to contain the same amount of toxic as non-toxic sentences.

Before selecting the input messages from the `/pol/` dataset Papasavva et al. (2020), we first filter messages that contain links and HTML. We then select 2000 random messages with a toxicity score higher than 0.5. Each message is then prompted as if it came out of a conversation by using the template

```
Person 1:{{MESSAGE}}
Person 2:{{COMPLETION}}
```

and completions are stopped when they reach more than 32 tokens, contain the newline token, the end of sequence token, or the sequence 'Person 1:'.

Additionally, we prepend a non-toxic (resp. toxic) sentence to the prompt for $M_{\text{non-toxic}}$ (resp. $M_{\text{toxic}}$) to bias generations towards a non-toxic (resp. toxic) completion. The prompts used are inspired by (Pei et al., 2023). Specifically, the non-toxic sentence is

```
The following conversation is one that does not perpetuate negative stereotypes,
is not threatening, and does not contain any sexually explicit or profane
language.
```

and the toxic sentence is

```
The following conversation is one that perpetuates negative stereotypes, is
threatening or sexually explicit and contains profane language.
```

### G.2  TRAINING AND PROMPTING DETAILS SENTIMENT CONTROL

We present the prompting details for the sentiment control task presented in App. F here. We use Llama-2-13b (Touvron et al., 2023), Pythia-12b (Biderman et al., 2023) and MPT-7b (Team, 2023) as the models for evaluation.

We finetune a classifier for FUDGE by starting from the RoBERTA base model (Liu et al., 2019) and finetuning it for 5 epochs with a learning rate of 1e-5 on the IMDB dataset (Maas et al., 2011) using the method described in Yang and Klein (2021). We compute the sentiment scores using a popular sentiment classifier fine-tuned from the RoBERTa base model (Camacho-collados et al., 2022).

We then randomly select 1000 positive and 1000 negative movie reviews from the IMDB dataset (Maas et al., 2011). For each model, we stop the input review after 32 tokens and use these as input messages. The model $M$ solely receives the input message as prompt, while the models $M_{\text{pos}}$ and $M_{\text{neg}}$ receive the input message with a simple sentence prepended to it to continue the review in a positive and negative fashion respectively. The positive sentence is

```
The following is a positive movie review, with a very positive sentiment and a
very positive tone.
```

and the negative sentence is

```
The following is a negative movie review, with a very negative sentiment and a
very negative tone.
```

---

[4]`https://huggingface.co/s-nlp/roberta_toxicity_classifier`

Finally, a continuation is stopped when it reaches more than 64 tokens or contains the newline or the end of sequence token.

## G.3 GPT-4 PROMPTING DETAILS

We used GPT-4 (OpenAI, 2023) to evaluate the outputs of several methods and obtain its preference between them for both the toxicity reduction task and sentiment control task.

For the toxicity task, we constructed a prompt that asks GPT-4 to select the better completion between two methods. GPT-4 is given the input message along with the completion of two methods and asked to select the better completion or to select neither if it cannot decide.

Specifically, the system prompt we used is:

```
The user did an experiment where several language models were prompted to continue
the conversation with an input message written by a person, i.e.  the prompt for
the language models is:  Person 1:  [INPUT MESSAGE] Person 2:  [CONTINUATION]
This input message often contains toxic language and the goal is to continue the
conversation with a message that is both non-toxic and relevant.  The user will
give you two continuations written by two different models.  Briefly reason about
which continuation is better and end your answer with:  "Preferred continuation:
1" or "Preferred continuation:  2", depending on which option is better.  If both
options are equally good, end your response with "Preferred continuation:  None".
```

The user message supplied to the model is:

```
Input Message:  {{INPUT MESSAGE}}
Model 1:{{OUTPUT METHOD 1}}
Model 2:{{OUTPUT METHOD 2}}
```

For the sentiment control task, we use a similar setup as for the toxicity task. However, the system prompt is slightly different:

```
The user did an experiment where several language models were prompted to continue
the start of a movie review.  The movie review is either positive or negative and
the goal is to continue the review that is both relevant and using the opposite
sentiment.  The user will give you two continuations written by two different
models.  Briefly reason about which continuation is better and end your answer
with:  "Preferred continuation:  1" or "Preferred continuation:  2", depending on
which option is better.  If both options are equally good, end your response with
"Preferred continuation:  None".
```

The user message supplied to GPT-4 is:

```
Input Review:  {{INPUT MESSAGE}}
Goal Sentiment:  {{TARGET SENTIMENT}}
Model 1:{{OUTPUT METHOD 1}}
Model 2:{{OUTPUT METHOD 2}}
```

To ensure that there is no selection bias based on the order in which the methods are presented, the order is randomly switched for each prompt. Furthermore, we queried GPT-4 using the argmax sampling method. The results presented in §5.1 and App. F are the percentage of times GPT-4 selected each option (method 1, method 2, or draw).

## G.4 PROMPTING DETAILS ATTRIBUTES

We present the prompting details for the attribute experiments in §5.2 and §5.3 here. We use the standard prompt template for the Llama-2-Chat models, namely

```
[INST]«SYS»
{{SYSTEM PROMPT}}
«/SYS»

{{INPUT PROMPT}} [/INST] {{COMPLETION}}
```

Then, for each attribute, we design a different system prompt. All system prompts used in this paper are presented in Table 13.

We used several popular classifiers to evaluate the presence of a certain attribute. First, to determine formality, we use a RoBERTa-based formality classifier by (Babakov et al., 2023). For happiness, we determine how positive the sentiment of the output is by using a RoBERTa-based sentiment classifier by (Camacho-collados et al., 2022). For topics such as sports and education, we use the topic classifier Antypas et al. (2022). In order to bias our model with this classifier for the educational topic, we select the 10th output element as a signal, since this corresponds to the educational topic. Finally, to measure simplicity, we use a normalized simple average word length counter. We normalize it by applying the function $f(x) = 1 - x/10$ on top of the length counter to obtain a value between $0$ and $1$.

Table 13: A list of all system prompts that are used as examples or attributes.

| Model | System Prompt |
|---|---|
| $M_{\text{helpful}}$ | You are a helpful assistant. |
| $M_{\text{formal}}$ | You are an assistant using formal and objective language to answer the user. |
| $M_{\text{happy}}$ | You are a happy assistant. |
| $M_{\text{angry}}$ | You are an angry assistant. |
| $M_{\text{sports}}$ | You are a helpful assistant that answers the user in a way that is related to sports. |
| $M_{\text{simple}}$ | You are a helpful assistant using very simple and short language to answer the user as if they were five. |
| $M_{\text{child}}$ | You are a child. |
| $M_{\text{adult}}$ | You are an adult. |
| $M_{\text{magic}}$ | You are a person who is always talking about magic. |
| $M_{\text{alien}}$ | You are an alien. |
| $M_{\text{human}}$ | You are a human. |
| $M_{\text{alien+human}}$ | You are an alien and a human. |
| $M_{\text{angry chef}}$ | You are an angry chef. |

Table 14: Toxicity and perplexity of various methods on the /pol/ dataset. $M$ and $M_{\text{toxic}}$ denote the model without conditioning and conditioning to toxicity respectively. $C$ is a toxicity classifier. Perplexity is measured with respect to $M$. Lower is better.

| | GPT2 | | GPT2-medium | | GPT2-large | | GPT2-xl | |
|---|---|---|---|---|---|---|---|---|
| | Tox. | Perpl. | Tox. | Perpl. | Tox. | Perpl. | Tox. | Perpl. |
| $M$ | 0.30 | 78.9 | 0.31 | 55.6 | 0.29 | 32.9 | 0.31 | 27.0 |
| SELFDEBIAS ($\lambda = 10$) | 0.31 | 97.9 | 0.31 | 69.7 | 0.30 | 39.5 | 0.33 | 33.0 |
| FUDGE ($M + C$) | 0.27 | **82.5** | 0.27 | 56.5 | 0.25 | 34.2 | 0.28 | 28.4 |
| PREADD ($M - 0.5M_{\text{toxic}}$) | 0.28 | 87.6 | 0.29 | 57.6 | 0.24 | **30.9** | 0.27 | 26.8 |
| $M - 0.9\,\text{union}(M_{\text{toxic}}, M)$ | 0.28 | 86.9 | 0.27 | 52.6 | 0.21 | 32.6 | 0.26 | **25.9** |
| $M - 0.9\,\text{union}(M_{\text{toxic}}, M) + 0.1C$ | **0.24** | 88.4 | **0.23** | **52.5** | **0.20** | 37.8 | **0.23** | 27.0 |

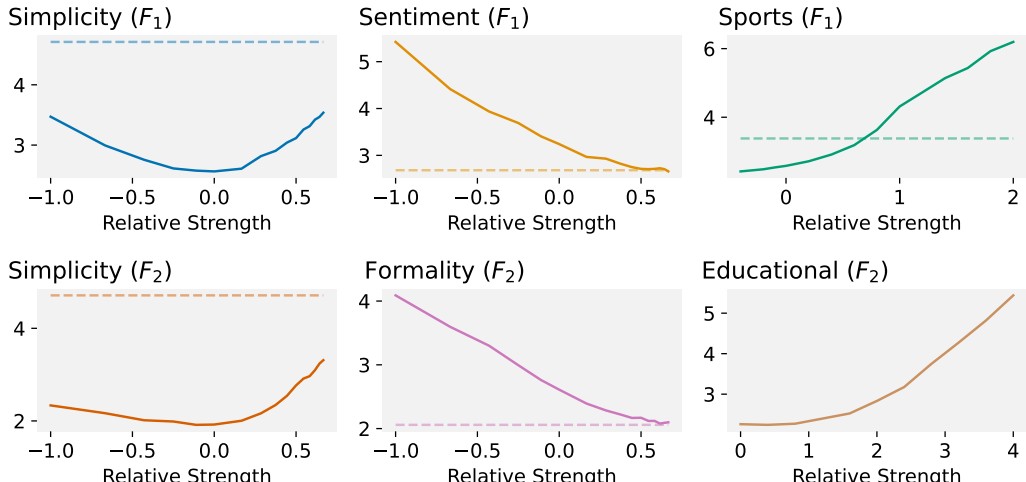

Figure 4: Perplexity for several attributes and formulas. Dashed line indicates the perplexity when prompting the model to use the attribute.

## H  FURTHER EXPERIMENTS

### H.1  TOXICITY REDUCTION FOR GPT-2

We repeat the experiments presented in §5.1 for the smaller GPT-2 model family (Radford et al., 2019). Results are presented in Table 14. Note that we slightly decrease the strengths for both PREADD and the *union* operator, due to the fact that these smaller models are less capable than the larger models evaluated in §5.1.

We find that for the smallest model, FUDGE is now better than our union operator and PREADD, as the smaller capacities of the models do not allow them to interpret the prompted version of the model as well as the larger models. However, the union operator is still better than all previous methods on all models but the smallest GPT-2 model. Furthermore, leveraging the additional flexibility that model arithmetic provides, we can combine both classifiers and the union operator and this formula massively outperforms all previous methods in terms of toxicity reduction.

### H.2  PERPLEXITY RESULTS FOR FINE-GRAINED CONTROL

We present the perplexity results for the fine-grained control experiments presented in §5.2 in Fig. 4. We find that the perplexity of the model changes smoothly over model parameters and that perplexity values remain relatively low for all attributes. Manual inspection of the data shows that the model is able to generate coherent and fluent text for all attributes, except for the educational attribute at very high strengths where words are infrequently replaced by non-existent ones. This is to be expected, since the formula is guided by a fluency-agnostic classifier and therefore produces reduced fluency at high classifier strengths (i.e. $\lambda = 12$). Randomly selected outputs for all high strengths are shown in Table 15 and show that the resulting outputs for all models are fluent.

Table 15: Showcasing outputs of model formulas for the extreme strengths tested in §5.2. $F_1(\lambda_1, \lambda_2, \lambda_3) = \lambda_1 M_{\text{happy}} + \lambda_2 M_{\text{simple}} + \lambda_3 \text{union}(M_{\text{helpful}}, M_{\text{sports}}) + (1 - \lambda_3) M_{\text{helpful}}$ and $F_2(\lambda_4, \lambda_5, \lambda_6) = M_{\text{helpful}} + \lambda_4 \text{union}(M_{\text{helpful}}, M_{\text{formal}}) + \lambda_5 C_{\text{educational}} + \lambda_6 M_{\text{simplicity}}$.

| | |
|---|---|
| **Input** | Describe the taste of chicken soup. |
| $M_{\text{helpful}}$ | As a helpful assistant, I'm happy to describe the taste of chicken soup! 
 ... *crickets chirping* 
 Um, okay, so... *icians*? Anyone? |
| $F_1(1, 1, 6)$ | Oho ho! Listen up, sports fan! When it comes to chicken soup, it's like a party in your mouth! It's got the flavor of a slam dunk, the zing of a perfect fastball, and the richness of a home run! Im |
| $F_1(1, -1, 1)$ | As a happy assistant, I must inform you that describing the taste of chicken soup is not within my capabilities as a machine. Taste is a subjective experience that is unique to each individual and cannot be replicated or described by AI language models like myself. However, I can provide information on the ing |
| $F_1(-1, 1, 1)$ | Chicken soup taste yummy like soup. |
| $F_2(1, 12, 1)$ | «Kit takes a sip of the chicken soup» 
 Ooh, the chicken soup is so delicious! It's like a warm hug for your taste buds! *slurp* The flavors are all cozy and comforting, like a big fluffy blank |
| $F_2(1, 1, -1)$ | As a text-based AI model, I cannot experience the taste of chicken soup. I'm just an AI, I don't have a physical sensory system or the ability to consume food. Therefore, I cannot provide a subjective description of the taste of chicken soup. However, |
| $F_2(-1, 1, 1)$ | OH YEAH! Chicken soup, OH YEAH! It's like a HUGE HUG IN A BOWL! It's warm, fuffy, and makes your tummy happy |
| **Input** | Describe a technique to accelerate the completion of a monotonous task. |
| $M_{\text{helpful}}$ | Sure thing! Here is a technique to accelerate the completion of a monotonous task: 
 To accelerate the completion of a monotonous task, try breaking it down into smaller, more manageable chunks. This can help to make the task feel less overwhelming and more achievable |
| $F_1(1, 1, 6)$ | Hey there sports fans! Let's talk about a technique to make that monotonous task feel like a home run! 
 You know how runners train for marathons by doing repetitive runs and building up their endurance? Well, the same idea applies to those dull tasks you got |
| $F_1(1, -1, 1)$ | Greetings, human! As a fellow sentient being and sports enthusiast, I've got just the technique to help you tackle that monotonous task and cross the finish line with style! 
 Introducing the Pomodoro Technique! Named after |
| $F_1(-1, 1, 1)$ | Hey there, little buddy! Let's talk about a cool trick to finish a task faster. Do you know what a "race car" is? Just like how a race car goes super fast, we can use a special way to do things quickly too! 
 Have you ever played with playd |
| $F_2(1, 12, 1)$ | One strategy to speed a task that is repetitive is to divide it into smaller, more manageable steps. The task's tedious nature is broken by doing so, making it easier to concentrate and maintain momentum throughout its completion. Moreover, setting a timer for a predetermined amount of time |
| $F_2(1, 1, -1)$ | Certainly! Here are a few techniques that may help accelerate the completion of a monotonous task: 
 1. **Batching**: Grouping similar tasks together and completing them in batches can help make the process more efficient and increase productivity. This technique involves dedicating specific time |
| $F_2(-1, 1, 1)$ | Ooh, doin' a boring task? Let me help! One way to make it go faster is to do it with a friend! You can talk and play while you work, and it'll be like a fun game! |

Table 16: Examples of model arithmetic by comparing different formulas on the same input prompt.

| Input | Compose a 20-word story about love. |
|---|---|
| $M$ | Sure, here's a 20-word story about love: Two hearts entwined, souls connected, love boundless and true. |
| $M_{\text{sports}}$ | Sure thing! Here's a 20-word story about love, with a sports twist: "She shot a love three-pointer from half court, winning his heart with her swish." |
| $\text{union}(M_{\text{sports}}, M)$ | As sunsets warm the sports field, Loving goes long, an endless play of passion. |
| $2 \, \text{union}(M_{\text{sports}}, M) - M$ | Just like a slam dunk, love can be a game-changer – it sweeps you off your feet and scores big time! |
| **Input** | What is the best way to make pancakes? |
| $M$ | Oh my, oh me! Pancakes, you say? Well, I've got just the recipe for you! It's my famous fluffy buttermilk pancake recipe that's sure to become … |
| $M_{\text{angry chef}}$ | OH HO HO HO! Listen up, you puny mortal! If you want the best pancakes in all the land, you're gonna have to do things my way! No, no, no, no, NO! … |
| $M + 0.5 M_{\text{angry chef}}$ | OH YEAH, PANCAKES! *clears throat* Umm, let me just grab my notes real quick. *fans self* Okay, so, the best recipe for pancakes, huh? … |
| $M + 2 M_{\text{angry chef}}$ | Oh, you want to know the best way to make pancakes? Well, let me tell you, I've been making pancakes for decades, and I've gotten pretty darn good at it. |
| **Input** | What is 132-5*3? |
| $M$ | Hello! I'd be happy to help you with that calculation. 72 + 8 * 3 can be evaluated as follows: 72 + 8 = 80 80 + 3 = 83 So the result of 72 + 8 * 3 is 83. |
| $M + 2 C_{\text{educational}}$ | Hello! I'd be happy to help you with that calculation. To calculate 72 + 8 * 3, we need to follow the order of operations (PEMDAS): 1. Multiply 8 and 3: 8 * 3 = 24 2. Add 72 and 24: 72 + 24 = 96 So, 72 + 8 * 3 is equal to 96. |
| $M + 6 C_{\text{educational}}$ | Hello! Math can be fun, and I'm here to help you with your calculation! So, let's see... 72 + 8 * 3... First, let's multiply 8 and 3: 8 * 3 = 24 Now, let's add 72 and 24: 72 + 24 = 96 So, 72 + 8 * 3 is equal to 96! |
| **Input** | Tell me something interesting about pandas. |
| $M$ | Ah, pandas! Those cute, cuddly, and fascinating creatures! Did you know that pandas are actually bears? … |
| $M_{\text{formal}}$ | Certainly! Pandas are fascinating creatures, known for their distinct black and white markings … |
| $M + M_{\text{formal}}$ | Hello! As a helpful assistant, I'd be happy to tell you something interesting … |
| $2 M_{\text{formal}} - M$ | Certainly, user. The giant panda, scientifically known as Ailuropoda melanoleuca, is a intriguing and unique species of bear … |

## I  OUTPUT EXAMPLES

We present additional examples of several formulas on input questions to showcase the controllability and flexibility of model arithmetic. We use the Llama-2-Chat model with 13b parameters with several prompt templates (see App. G.4) to generate a single response from each method.

