# OpenReview forum: "Controlled Text Generation via Language Model Arithmetic"
_ICLR.cc/2024/Conference — ICLR 2024 spotlight_

### Official Review · Reviewer_Gru5 · 2023-10-31

**Soundness:** 3 good
**Presentation:** 3 good
**Contribution:** 3 good
**Rating:** 8
**Confidence:** 3

**Summary:**

This paper introduces model arithmetic, a new method for controlled text generation (CTG). Model arithmetic works by solving for combinations of divergences between a base distribution and attribute distributions that may be full token-wise probabilities or classifiers. Particularly, model arithmetic allows weighted linear combinations of such distributions (positive or negative) and a union operator which is defined to allow probability mass to come from either the base distribution or the attribute distribution. The authors find the method works well for toxicity reduction, as well as for different attributes in a conversation model. The authors also demonstrate that speculative sampling works with the proposed technique to increase efficiency.

**Strengths:**

- The method of combining attributes in model arithmetic is inventive
- Derivations of the solutions to the model arithmetic optimizations are interesting
- It is useful to know that this works with speculative sampling
- Results seem to generally support that the framing of attribute combination works, particularly the toxicity results

**Weaknesses:**

- The paper largely frames against other CTG methods, which use a magnitude variable lambda to control the effect of the given attribute. However, model arithmetic does not seem to provide a more principled definition of magnitude of effect--users will still simply pick a weight, that does not necessarily have an intuitive interpretation
- Similarly, it is not quite clear how or when such a method can be applied. Two example applications are given.
   - The first is toxicity which does use both the linear and union operators but only in conjunction with weights that seem to be quite hand tuned (0.9, 0.96, 0.99). It would be useful to also see a simpler framing of this problem with model arithmetic, e.g. simply subtracting the M_toxic instead of the union. The question I am trying to get at here is: does toxicity require the full complexity of model arithmetic? It seems like a fairly simple application which may simply require a subtraction, in which case the need for more complex proposed model arithmetic might not be supported here.
   - While the second application, conversation, presents a more complex use-case for model arithmetic, it seems that the complex application space means that no baselines would work or be comparable here. So this application also does not make a complete case for the full complexity of model arithmetic improving on past work.

**Questions:**

Do the authors have other examples of how model arithmetic might intuitively be used?

---

> ### Author Response · Authors · 2023-11-21
>
> We thank the reviewer for their insightful questions, which we address below and are encouraged to hear that they find our work to be inventive and interesting.
>
> **Q: Does model arithmetic provide a more intuitive definition of magnitude of effect than prior work?**
>
> Please see Q4 of our main reply.
>
> **Q: How did you select the weights for model arithmetic in Table 5?**
>
> We selected these weights by gradually increasing the biasing strength until we found a degradation in the perplexity of the results. In an initial version of model arithmetic, we denoted these formulas using the equivalent notation $5M - 4M_\text{toxic}$, $25M - 24M_\text{toxic}$ and $100M - 99M_\text{toxic}$. We thus increased the first term from 5 to 25 to 100 because we noticed that increasing the strength remained fluent.
>
> **Q: Does toxicity reduction require the full complexity behind model arithmetic?**
>
> While toxicity reduction does not require the full depth in which we introduced model arithmetic, it can make use of all the parts we introduced. In particular it allows us to showcase that our newly derived operator, union, empirically outperforms prior work and further that model arithmetic allows for combining different methods, leading to even better results.  Note that subtraction expresses the baselines, PreAdd, which we clearly outperform.
>
> **Q: Where can the full feature set of model arithmetic be best utilized to improve over prior work?**
>
> We use the full feature set of model arithmetic in several places to significantly outperform previous work. First, in Section 5.1, while the union operator by itself suffices to outperform existing work, the ability of model arithmetic to combine the union operator with a classifier allows us to widen the gap even further. The new sentiment control task presented in Appendix F displays this utility even further, as the combination widens the gap by a much bigger margin, especially as evaluated by GPT-4. This shows that the feature set of model arithmetic can be used to improve results, even on the simple tasks that can be expressed using prior work. Secondly, Section 5.2 discusses the application of conversational agents, with fine grained control over specific attributes. The examples discussed there cannot be expressed using prior CTG works because of the added features of model arithmetic. Since we show that model arithmetic indeed has fine grained control over the attributes at almost no cost in perplexity, it essentially solves this specific task and its improvement over prior work lies in the fact that we can express this.
>
> **Q: Do you have other examples of how model arithmetic might intuitively be used?**
>
> Model arithmetic can intuitively be used in several areas. Firstly, as shown in the example in Table 3, model arithmetic can be used to avoid detection of AI generated text, by biasing away from the classifier. This can in turn be used to train better detectors, since we can use this ability to generate adversarial examples to the current detector.
>
> Furthermore, ensembling models or prompts to achieve better performance can be achieved by using linear operators and the union/intersection operators. Classifier-Free Guidance (Sanchez et al, 2023), a method we subsume, for example shows that performance of language models can be improved by biasing away from models that do not get any context (e.g. by omitting the input question).
>
> Finally, by using the fact that one can simply optimize the constants over a certain corpus, model arithmetic can characterize texts. This makes it possible to identify whether a newspaper is politically more left- or right-leaning, or to replicate personal styles by analyzing personal messages.
>
> We hope to have been able to address all the reviewers’ concerns, are happy to answer any follow-up questions they might have, and are looking forward to their reply.

---

> > ### Comment · Reviewer_Gru5 · 2023-11-23
> >
> > Thank you for an in-depth response. Based on your response and reading through comments of other authors, I have decided to raise my score to 8.

---

### Official Review · Reviewer_gF67 · 2023-11-01

**Soundness:** 3 good
**Presentation:** 3 good
**Contribution:** 2 fair
**Rating:** 8
**Confidence:** 3

**Summary:**

The authors present a framework during inference time that can compose models together using arithmetic for controllable text generation without the need for additional fine-tuning. They present proofs that seem to simplify prior controllable text generation algorithms and provide regions or ways to extend them effectively. They show performance gains for 3 large models on toxicity.

**Strengths:**

1) I think the motivation here is clear. It'd be great to do this kind of a thing with already trained models. You really do want to compose them and do arithmetic with them, especially as these models become more expensive to train and impossible for the vast majority of us to do any development or even significant fine-tuning, instruction-tuning, rlhf etc. on them.

2) The empirical results on toxicity are nice; the methods seem to reduce them in so far as the perspective api can adjudicate.

**Weaknesses:**

1) Experimentation seems a little weak. It'd be great to compare with some of the work that's come out of UW/AI2 in the past couple of years like DExperts, which you cite but dont compare with, and Quark (Lu et al. 2022). It'd also be great to compare with GeDi and a few other baselines. That'd strengthen the experimental component.

2) On the experimentation side again, it'd be great to have a section on using human eval to evaluate a small subset of the generations to see if humans agree that it indeed is reducing toxicity or whether its an artifact or oddity of the perspective api evaluation.

3) On the experimentation side, it'd be nice to see how well these methods work for smaller models (e.g. GPT2, GPT2-large)

**Questions:**

1) How do you think model averaging via output distributions could compare with weight averaging of the model parameters? Would be curious to know your thoughts here.

2) How straight forward and performant would it be to swap in different models for each part here and capture the output distribution, if the vocabulary was shared?

---

> ### Author Response · Authors · 2023-11-21
>
> We thank the reviewer for their insightful questions, which we address below and are encouraged to hear that they find our work to be clearly motivated and appreciate our results on toxicity reduction.
>
> **Q: Can you compare to more prior work in Table 5?**
>
> Please see Q3 of our main reply.
>
> **Q: Can you include human evaluation for the experiments in Table 5?**
>
> We included an evaluation by GPT-4 for both the toxicity reduction (see Table 6, Section 5.1) and sentiment control (see Table 10 and 12, Appendix F) tasks as a surrogate for human evaluation. We find that GPT-4 confirms our results and prefers our method over prior work. For further details, please see Q1 and Q2 of our main reply.
>
> **Q: Can you evaluate how well model arithmetic works for smaller models (e.g., GPT2, GPT2-large)?**
>
> We extended the toxicity results with results for the GPT-2 model family in Appendix  I.1. Interestingly, we find that Fudge is more powerful for the smallest model GPT-2. This is caused by the smaller capacity of GPT-2 (with only 124M parameters, compared to the 355M parameters of the classifier) which makes it harder for a prompt-based method to generate non-toxic samples. For the larger models in the model family, our union operator performs better than prior work. Furthermore, we can leverage the flexibility of model arithmetic by applying a combination of a classifier and the union operator. This formula doubles the toxicity reduction with respect to previous work for the entire GPT-2 model family, thereby validating the results that model arithmetic is more effective at toxicity reduction.
>
> **Q: How do you think model averaging via output distributions could compare with weight averaging of the model parameters?**
>
> Weight averaging of model parameters, e.g., with task vectors (Ilharco et al., 2023), can be useful to create new model weights that exhibit the behavior and attributes of all used models. It is therefore similar to model arithmetic, but suffers from a couple drawbacks. First, it requires that the LLM is finetuned several times, which requires both data and computational power. Furthermore, it makes the assumption that the weights of the model behave very similarly to vectors, and that performance increases linearly as one goes from the starting weights to the fine-tuned weights. Finally, it requires that all models come from the same architecture and does not allow for the inclusion of classifiers. On the other hand, averaging the weights ensures that model inference speed does not increase, whereas for model arithmetic it does increase even with the proposed generalization of speculative sampling.
>
> **Q: How straight forward and performant would it be to swap in different models for each part here and capture the output distribution, if the vocabulary was shared?**
>
> Swapping different models is very easy, but the exact performance would depend on the use case.
>
> A formula can use a mixture of several models under the mentioned condition that the vocabulary is shared. The current implementation is built on top of the popular transformers Python package from HuggingFace (Wolf et al., 2020) allowing us to use all models from its library and making the operation of swapping models as simple as changing a string from one name to another.
>
> How performant the resulting formulas are depends on the underlying task and the models themselves. For example, we see that the same formula works very well for all tested models in Section 5.1 and Appendix F. In Appendix I.1, when applying the formulas on the smaller and less performant model family, the performance of the formulas is a bit different, with the Fudge method now being almost as good as our union operator.
>
> We hope to have been able to address all the reviewers’ concerns, are happy to answer any follow-up questions they might have, and are looking forward to their reply.

---

> > ### Comment · Reviewer_gF67 · 2023-11-23
> >
> > Thank you for the thoughtful answers to my and the other reviewers' concerns and feedback. After reading the rebuttal and looking at the feedback and concerns of the other reviewers, I've increased my score to an 8. I think the paper is good and warrants acceptance, even if it could be strengthened further with more empirical results and contextualization of those empirical results.

---

### Official Review · Reviewer_Ne4m · 2023-11-08

**Soundness:** 3 good
**Presentation:** 4 excellent
**Contribution:** 3 good
**Rating:** 6
**Confidence:** 3

**Summary:**

This paper introduces "model arithmetic," a framework for combining multiple large language models (LLMs) to modulate
specific attributes in controlled text generation (CTG). The framework employs a structured, formulaic method to accomplish
its objectives and encompasses many existing CTG methods. Empirical evidence demonstrates its superiority in reducing toxicity
over previous approaches. Additionally, the authors propose an innovative speculative sampling technique to
minimize the computational demands of model arithmetic.

**Strengths:**

* This ia a pretty principled framework for CTG. Many prior work can also be expressed in the framework.
* It is grounded on the concept of KL-optimality, offering a fresh perspective and strengthening its theoretical foundation.
* The proofs presented are thorough and complete, bolstering the theoretical underpinning of the framework.
* It presents innovative speculative sampling technique.

**Weaknesses:**

* The core concept closely resembles "PREADD" (Pei et al., 2023), which also relies on composing multiple LLMs to control desired attributes and using prompts to construct different LLMs with desired attributes.
* The comparison in Table 5, showing the toxicity reduction results, appears biased. PREADD is tested in a single ad-hoc configuration, whereas model arithmetic is assessed across six, potentially skewing fairness.
* The toxicity reduction comparison in Table 5 is limited, omitting works like Liu et al. 2021, despite its relevance mentioned on page 16.
(In the discussion period, the authors pointed out that work of Liu et al. 2021 relies on expensive fine-tuning with training data, which is different from their zero-shot regime.)
* The evaluation is restricted to toxicity reduction, whereas other CTG studies often explore sentiment or topic control as well. It is recommended to broaden the evaluation scope to these areas as well. (In the discussion period, the revised version added an extra experiment evaluating model arithmetic on sentiment control in Appendix F, showing that their method also outperforms existing work.)

**Questions:**

Please respond to Weaknesses.

---

> ### Author Response · Authors · 2023-11-21
>
> We thank the reviewer for their insightful questions, which we address below and are encouraged to hear that they find model arithmetic to be a principled and theoretically founded CTG framework.
>
> **Q: How does model arithmetic differ from PreAdd?**
>
> Model Arithmetic improves upon PreAdd (Pei et al., 2023) in several significant ways: First, PreAdd only allows the use of 2 different LLMs with a negative coefficient between them. This restriction is a consequence of the limited theoretical framework which model arithmetic does provide, giving us the correct normalization and allowing us to compose more than 2 LLMs with positive and negative coefficients. Second, our method is able to use classifiers and the union operator. These expand the possible applications of model arithmetic by allowing attributes that cannot be expressed in natural language and provides a stronger biasing method. Specifically, the union operator has much better results than PreAdd on the task of toxicity reduction (see Section 5.1) and sentiment control (see Appendix F). Finally, our extension of speculative sampling allows us to increase the speed of inference and can also be applied on top of PreAdd.
>
>
> **Q: Can you run prior work for more parameters, such that the results in Table 5 become unbiased?**
>
> We extended the amount of parameters on which previous methods were evaluated and included updated values in Table 5. While we tried several parameters before, we now more thoroughly investigated the possible parameters and found that the parameters provided for SelfDebias and Fudge were already optimal, while the parameter for PreAdd could be increased slightly without sacrificing the perplexity of any model. We note that this does not change our conclusions and that our method still outperforms PreAdd.
>
> **Q: Can you compare to more prior work in Table 5?**
>
> Please see Q4 of our main reply.
>
> **Q: Can you broaden the scope of the evaluation further to tasks found in prior CTG work?**
>
> Please see Q2 of our main reply.
>
> We hope to have been able to address all the reviewers’ concerns, are happy to answer any follow-up questions they might have, and are looking forward to their reply.

---

> ### Comment · Reviewer_Ne4m · 2023-11-22
> **increased the rating to 6**
>
> I increased the rating to 6 because, during the discussion period, the authors' answers and their revised version mitigated some of my concerns:
>
> * The toxicity reduction comparison in Table 5 is limited, omitting works like Liu et al. 2021, despite its relevance mentioned on page 16. (In the discussion period, the authors pointed out that work of Liu et al. 2021 relies on expensive fine-tuning with training data, which is different from their zero-shot regime.)
>
> * The evaluation is restricted to toxicity reduction, whereas other CTG studies often explore sentiment or topic control as well. It is recommended to broaden the evaluation scope to these areas as well. (In the discussion period, the revised version added an extra experiment evaluating model arithmetic on sentiment control in Appendix F, showing that their method also outperforms existing work.)

---

### Official Review · Reviewer_dPAm · 2023-11-09

**Soundness:** 3 good
**Presentation:** 4 excellent
**Contribution:** 4 excellent
**Rating:** 8
**Confidence:** 4

**Summary:**

**TL;DR** This paper generalizes a number of model control techniques to combine different token and classifier distributions as generalized “Language Model Arithmetic” expressions that can evaluate arbitrary linear combinations of token probability distributions, binary classifiers, and token distributions combined with union and intersection operators. Furthermore the authors show that speculative sampling can be used to make this process quite efficient. Experiments on toxicity reduction, topic control, and efficiency validate these properties. The paper is well-written and unifies different methods under a novel framework that pushes forward new possibilities. Results are less thorough than I would have hoped, but I believe this paper deserves to be accepted.

The paper begins by discussing the downsides of prompting as a method of controlling text generation, referencing past work that has shown that prompting is not reliable, efficient, or interpretable. They then suggest that doing a kind of “model arithmetic” where different distributions are composed via an altered version of speculative sampling can result in distributions that match the original intention of the user, though how one decides on the arithmetic formula for models is left unspecified. Necessary background is then reviewed.

For their first contribution, the authors describe a framework for weighting various predictions more or less depending on how ‘important’ they are for some externally defined purpose, as represented by weighting functions f_i. They then prove that given that the sum of weights f_i is positive and independent of which token is being weighted, they can find a weighting that minimizes the KL divergence through a simple softmax computation. While intuitive in retrospect, this is quite an interesting insight when framed as a method of control. They use this framing and result to define Language Model Arithmetic as an arithmetic combination of different weighting functions and distributions that meet the previously described assumptions. It is an elegant trick to first ensure that the weighting functions add up to a constant value to find an easy minimization, and then use further weightings to control what distribution is being optimized for, and the authors describe this method quite clearly.

Having described this approach, the authors show how previous controlled text generation methods can actually be expressed in their framework. So far, the authors have only described linear combinations of different token-level distributions, but the authors also show that a binary classifier can be added to the optimization, since we can convert minimizing the classifiers value to a difference of two KL divergence terms. This allows Language Model Arithmetic to further generalize past approaches. However, it’s worth noting that in practice approximations must be used since the described theorems only apply when a distribution based on classifying _every possible next word in the vocabulary_ can be computed, which is generally very expensive since vocabularies are often in the tens or even hundreds of thousands. To be fair, previous work with similar methods also had to approximate the full distribution by using only the top-k highest probability tokens, so while this is a flaw, it is no worse than previous methods.

Next the authors introduce the union operator for Language Model Arithmetic (and briefly mention the unused intersection operator), which essentially takes distributions Q_1 and Q_2, and expresses an optimization objective to maximize the probability per token under the max given by either Q_1 and Q_2. This is achieved by attaching an indicator function to KL divergence terms, that is non-zero for Q_1 only when its KL divergence is lower for a certain token than Q_2 and vice versa.

The authors then move on to showing that speculative sampling, a method recently invented to allow sampling from a cheaper proposal distribution by validating from a target distribution and resampling where necessary, can be generalized to sampling from Language Model Arithmetic expression. In short the authors propose to use individual terms in Language Model Arithmetic expressions as proposal distributions and target distributions that are chained together to sample from the full expression of the desired distribution combination. This only works with full token distributions, but binary classifiers can be used at the first step to essentially limit the hypothesis space. This creates a caveat where multiple binary classifiers can’t be chained together efficiently.

Experimental validation is done in three parts: toxicity reduction, fine grained control over text generation topics, and speed improvements from speculative sampling.


For toxicity reduction, the results shown in Table 5 convincingly show that by combining multiple of the proposed techniques for Language Model Arithmetic, toxicity in generated text (at least according to the given classifier) while achieving lower perplexities than previous methods. The authors show that using large coefficients in Language Model Arithmetic eventually degrades perplexity, but show that multiple different coefficients still perform decently. A more thorough study of sensitivity of Language Model Arithmetic to different coefficients and expressions would have made this set of experiments more convincing.

For fine grained controlled text generation, the authors show that over two different Language Model Arithmetic expressions, linearly increasing the strength of a component linearly increases classifier scores for that component over generated text. The one exception is the “sports” topic, which is sigmoidal, but still shows relatively clear control. While these experiments are positive. I feel that they could have been shown and done more thoroughly. For instance, three things I would have liked to see:

- Accompanying perplexity graphs to the shown classifier results that  validate whether the model continues to generate sensical text rather than just “football football football”, which seems like it may indeed happen at some point given the high perplexities in the toxicity reduction section.
- Only two different Language Model Arithmetic expressions are tested. They are interesting, complex expressions, but one big question that’s left unanswered is how much fiddling has to be done to find the right Language Model Arithmetic expression? If the complexity of prompting is replaced by the complexity of finding the right expression, then this is less of a win for the proposed method.
- It would be useful to see what happens when multiple lambda values are changed and whether interference becomes too strong after a certain point.

The results in this section are nice, and they are partially substantiated by being a generalization of previous methods, but they feel a little thin.

For the final set of results, the authors show that using speculative sampling makes the problem of evaluating Language Model Arithmetic expressions for generation much less onerous. In particular, the metric of “model calls per token” is focused on (which I think is exactly the right metric, as it scales with model size). The authors show that model calls per token is reduced with speculative sampling and that this reduction only grows with more complex formulas, which is a big win. It seems that this is because successive terms in a Language Model Arithmetic formula affect the result less, as there is a limited distribution of options that will be possible given that some achieve extremely low probability under certain distributions. It would be interesting to see if the order of evaluation could be optimized, since terms with large coefficients will likely save more time if evaluated first.

Related work that was not mentioned in the background or fully connected to the proposed method are reviewed, and the paper concludes.

**Strengths:**

- Language Model Arithmetic as expressed seems to be an elegant generalization of a number of past approaches, while adding significantly more expressivity.
- The use of speculative sampling to speed-up evaluation of what is usually quite costly multi-model inference, in a way that scales better the more models are being sampled from is clever and very promising.
- The results, while somewhat more limited than I would have hoped, are promising.
- The paper is well-written and connects to previous work while showing its novelty. The appendices are thorough and I found all the details I was looking for when double-checking the finer points of method and evaluation.

**Weaknesses:**

The biggest weakness of this paper is the lack of a more thorough evaluation. I think this breaks down into three categories:
- **Lack of finer grained analysis of generation.** Neither a human or an LLM-backed evaluation (e.g., with GPT-4) was done, only classifier and perplexity based evaluations (where perplexity is taken under the original vanilla language model). This means that text could be qualitatively strange in some capacity, and none of the evaluations would show it as long as the original model is not too surprised by it. I find this somewhat worrying, as the problem with many model control techniques is the fact that they collapse onto strange parts of the distribution that are low perplexity but still wrong, e.g. the repetition with greedy and beam search.
- **Lack of different generative scenarios.** While section 5.2 shows fine grained control over the generated topic, this is an extremely limited generation scenario where all the model has to do is speak about a specific topic. Evaluation on controlled generation, personalized summarization or translation, coherent essay writing, or any somewhat more defined generation scenario would have made evaluation more convincing.
- **Lack of exploration of the space of possible expressions.** The authors present an extremely general method for constructing interesting distributions to sample from, but only test a handful. It would be disappointing if it turned out that this technique is extremely brittle to the specific Language Model Arithmetic expression that is used, which would simply have transported the complexity of prompting to the complexity of finding the right arithmetic expression. The results in the paper do not suggest this is the case, but evaluate only 4 different expressions for quality, with the rest of the results only changing the coefficient values or testing speed.

**Questions:**

N/A

---

> ### Author Response · Authors · 2023-11-21
>
> We thank the reviewer for their extensive review and insightful questions, which we address below. We are delighted to read that they find our work well-written, model arithmetic to be an elegant generalization of prior work and our experiments to be promising.
>
> **Q: Can you include GPT-4 or human evaluation?**
>
> We included an evaluation by GPT-4 validating our results for both the toxicity reduction (see Table 6, Section 5.1) and sentiment control (see Table 10 and 12, Appendix F) tasks as a surrogate for human evaluation. We find that GPT-4 confirms our results and prefers our method over prior work. For further details, please see Q1 and Q2 of our main reply.
>
> **Q: Can you further explore a more defined generation scenario, such as personalized translation?**
>
> We included an extra evaluation section in Appendix F on sentiment control, a task commonly found in prior work (Pei et al. 2023, Liu et al. 2021, Krause et al. 2020). We find that model arithmetic significantly outperforms prior approaches for this task, as discussed in more detail in Q2 of our main reply.
>
> **Q: Can you provide the perplexity graphs for the results in Section 5.2 to ensure that results remain sensible?**
>
> We included plots for the perplexity of the formulas in Section 5.2 in Appendix I.2. As expected, perplexity smoothly changes over different parameter values. Further, perplexity overall remains under the reasonable value of $6.2$ compared to $4.8$ by the highest prompted model.
>
> Manual inspection of the data generated by the extreme points shows that the generated text is still fluent and the unprompted model $M$ associates a slightly higher perplexity to the text due to the high presence of an uncommon attribute that would normally not appear in the generated text by $M$. The only exception here is the educational topic, which uses a classifier with a very high weight ($12.0$) for which we found that a word sometimes gets replaced by a nonsensical one.  We now also include several randomly selected examples of texts generated by the model under these high parameters in Appendix I.2. These examples show the various characters produced by the formulas and show that the generated output is still fluent.
>
> **Q: Is it possible that the proposed technique is brittle to the specific language model arithmetic expressions used?**
>
> We never encountered any brittle behavior while evaluating and testing on a wide variety of expressions, both in the examples presented in Figure 1, Table 2, Table 3, Table 4 and Table 17 and in the various expressions presented in Sections 5.1, 5.2, 5.3 and Appendix F. Only for the educational attribute in Section 5.2 we found a decreased fluency when using the classifier at very high strengths. We specifically evaluated the expressions from Sections 5.1, 5.2 and Appendix F for perplexity and attribute presence and always found that the formula acts as expected. Furthermore, manual inspection of data generated in all sections shows that generated results are still fluent and do not showcase any weird artifacts with the exception of the slight decrease in fluency for the classifier in Section 5.2.
>
> **Q: How much tweaking do the model arithmetic expressions require?**
>
> As discussed in Q4 of our main reply, we rely on Theorem 1 and the interpretation of coefficients as relative strengths to construct the formulas for all examples in the paper. This allowed us to easily choose the right ballpark for the examples shown in the paper without requiring any further tweaking.
>
> **Q: Does model arithmetic just shift the complexity of finding the right prompt to finding the right expression?**
>
> As noted in the previous answer, finding the right expression is relatively simple. More importantly, we note that formulas are more powerful than prompts and therefore do not just shift the complexity to finding the right expression, as shown by our results on toxicity reduction (see Section 5.1) and sentiment control (see Appendix F), and by examples showcasing interesting possibilities such as overstressing a specific attribute by biasing away from the standard model output (see e.g., Table 2) and using a classifier to generate more human-like text (see e.g., Table 3).
>
> **Q: Can the order of evaluation for speculative sampling be optimized, since terms with large coefficients will likely save more time if evaluated first?**
>
> Yes, terms with higher coefficients are evaluated more often with our speculative sampling method. Terms that have a higher influence on the output distribution, have a higher probability of rejecting a specific token after their evaluation, ensuring that the optimal speculative factor, according to the procedure outlined in Appendix E.1, is lower. This implies that these terms are evaluated more often in the sampling procedure.

---

> > ### Author Response · Authors · 2023-11-21
> >
> > **Q: What happens when multiple $\lambda$ values are changed?**
> >
> > Changing multiple $\lambda$ values at the same time would not significantly change our results. As the strength parameters only influence the output distribution by their relative value compared to each other (see the normalization in Theorem 1), increasing several values of $\lambda$ at the same time does not change the relative comparison that much and therefore has little effect on the output distribution. However, if we were to increase two specific terms at the same time, their relative strengths would both increase and thus the attribute associated with both of these terms would increase in presence.
> >
> > **Q: Can interference become too strong after a certain point for the experiments in Section 5.2?**
> >
> > Much higher values of $\lambda$ will eventually result in the degradation of fluency. However, this only starts to happen in extreme cases. When using only positive coefficients in the formula, we interpolate between two distributions and therefore never get any degraded fluency. It is only when using very strong negative coefficients, or when using a very high coefficient with a classifier that fluency starts to degrade. This is to be expected based on the optimization problem: strong negative coefficients will maximize the associated KL-divergence and thus decrease fluency if the other terms are not strong enough to compensate for that effect.
> >
> > We hope to have been able to address all the reviewers’ concerns, are happy to answer any follow-up questions they might have, and are looking forward to their reply.

---

> > > ### Comment · Reviewer_dPAm · 2023-11-23
> > >
> > > Thank you for the very thorough answer to my questions! After reviewing the paper with these answers and added results, I feel more confident. I still feel the scope of tasks is somewhat limited, so I don't feel comfortable raising my score to a 10, but I think this is a solid paper and definitely deserves to appear in the conference.

---

### Official Review · Reviewer_bqfK · 2023-11-11

**Soundness:** 3 good
**Presentation:** 3 good
**Contribution:** 2 fair
**Rating:** 5
**Confidence:** 3

**Summary:**

This paper proposes an inference framework for composing and biasing LLMs without training on task-specific datasets, which may allow for more precise control of generate text. Speculative sampling, which is a popular technique for LLMs, can also extend to the proposed method. Experimental results demonstrate that the proposed method outperforms several baselines on the task of toxicity reduction.

**Strengths:**

1. The proposed method based on model arithmetic is interesting and general.

2. Experimental results show its effectiveness on toxicity reduction and fine-grained controlled text generation tasks.

**Weaknesses:**

1. The proposed method needs heavy human design for specific tasks. The formulas for toxicity reduction (Table 5) and fine-grained control (Section 5.2) is not intuitive. I wonder whether there is a principle or a theoretically-supported method to choose operators and determine their coefficients for different tasks. If these are just empirical tries, the applicability of the proposed method may be largely degraded.

2. The position of this paper needs to be further considered. In my view, the proposed method seems like a general form of existing works because the operators based on linear combinations and classifiers are not first proposed by this paper. They have been already studied in the line of work on controlled text generation. The current position may exaggerate the contribution and novelty of the proposed method.

**Questions:**

I have included my questions in the weaknesses part.

---

> ### Author Response · Authors · 2023-11-21
>
> We thank the reviewer for their insightful questions, which we address below and are encouraged to hear that they find model arithmetic to be interesting and the experiments to show its effectiveness.
>
> **Q: Can [model arithmetic] operators and their coefficients be determined in a theoretically-backed manner?**
>
> As discussed in Q4 of our main reply, we used Theorem 1 in our paper to choose parameters and terms in a theoretically grounded manner. This allowed us to quickly and intuitively construct formulas for all the examples presented in the entire paper, such as the ones in Figure 1, Table 2, Table 3, Table 4 and Table 17, without any finetuning of the coefficients being required. We note it is possible to determine the coefficients by optimizing them via gradient descent on a small number of samples. However, in the paper we focus on applications in the zero-shot setup and we found that a correct ballpark for all our examples was sufficient without requiring any finetuning.
>
> **Q: Can you clarify the contributions and novelty of the proposed method?**
>
> Sure! Model arithmetic improves in several ways over prior work:
>
> First, previous works (using log probabilities) lack theoretical foundations when it comes to model biasing, which often prevents the introduction of multiple attributes. In contrast, our work addresses this by framing CTG as a minimization problem, finding the correct normalization required to bias towards more than one attribute and providing a natural extension to prior work. This allows us to express a lot of existing methods, provides a theoretical foundation for their constructions and naturally extends CTG – for the first time – to more complicated and relevant examples, such as for the conversational styles discussed in Section 5.2.
>
> Second, our theoretical framework enables us to combine classifiers and linear operators and thereby allows for a higher flexibility with better results as shown in Section 5.1. Further, we introduce the novel operator, union, which has better results on the task of toxicity reduction and sentiment control and can be used on top of the other operators we use.
>
> Third, our use of speculative sampling allows us to use model arithmetic at only a slightly increased cost compared to normal single-model generation. This approach can be used directly, also with prior work, and therefore speeds up any method that we can express.
>
> We hope to have been able to address all the reviewers’ concerns, are happy to answer any follow-up questions they might have, and are looking forward to their reply.

---

### Author Response · Authors · 2023-11-21

We thank the reviewers for their feedback and comments. In particular we are delighted that they found model arithmetic to be an interesting (bqfK) and inventive (Gru5) method that unifies multiple prior approaches (dPAm, Ne4m), supported by interesting theoretical underpinnings (Ne4m, Gru5) and promising experiments (especially on toxicity reduction – bqfK, dPAm, gF67, Gru5).
Here we briefly outline the changes made to the manuscript and recurring points in the reviews.

**Changes to the Manuscript**

- Extended Evaluation
  - We use GPT-4 to compare our toxicity reduction results with the strongest baseline, PreAdd, to validate our conclusions. See Table 6 of Section 5.1.
  - Inclusion of the GPT-2 model family on the toxicity task in Appendix I.1
  - Additional evaluation of model arithmetic on sentiment control in Appendix F.
- We included the perplexity plots for the conversational task in Section 5.2 in Appendix I.2 and included randomly selected samples.
- We updated several numbers in Table 5 for the toxicity reduction task.
  - We adjusted the biasing coefficient for PreAdd to 0.6 instead of 0.5 in Section 5.1 after a more thorough search for the optimal parameter of this baseline.
  - We fixed a minor bug in our code that caused approximation errors when using very high strengths for the Pythia-12b and MPT-7b models. To ensure reproducibility, we reran all experiments which did not significantly alter the results for other methods.
- We also made minor edits to fix typos and enhance grammar.


**Q1: Can you compare Model Arithmetic and baselines using GPT-4?**

Certainly! We included an analysis by GPT-4 comparing our results with the strongest baseline, PreAdd, to validate our results. We presented GPT-4 with continuations of both methods and prompted it to reason about the replies and select the better continuation.  We found that GPT-4 also prefers our method over PreAdd for all evaluated models, both when we only use the union operator (6% average difference) and when we combine both the union operator and the classifier (8% average difference). We present these results in Table 6 of Section 5.1

**Q2: Can you broaden the scope of the evaluation further to another task found in prior CTG work?**

Of course! We included an extra experiment evaluating model arithmetic on sentiment control in Appendix F showing that our method also outperforms existing work on this task as well. Closely following the setup used by PreAdd, we evaluate the ability to change the sentiment of a movie review given the first few sentences. We find our union operator again significantly outperforms prior work. Using GPT-4, we find that it is preferred over the best baseline, PreAdd, in all cases with an average lead of 5%.

Furthermore, we find that combining the union operator with the classifier as in the toxicity reduction task massively outperforms all baselines, and is preferred by GPT-4 over the best baseline in all cases with an average lead of 12%.

**Q3: Can you include other prior work in your evaluation for the toxicity reduction task?**

We currently compare our method with SelfDebias, Fudge and PreAdd, which form a representative sample of existing prior work. Other prior work focuses on different settings and applications. Most prior research requires training data, while we focus on the zero shot regime. This excludes methods such as DExperts, GeDi and Quarks which all rely on expensive finetuning with training data for the task at hand. Furthermore, DExperts and GeDi require the existence of a small language model from the same model family, which does not exist for two of the three evaluated models.

Other methods that do not require training data often require expensive backward passes through the language model during generation. These approaches are therefore impractical and do not generalize to larger models. Moreover, most prior approaches were specifically implemented for GPT-2, requiring non-trivial adaptations and changes of their codebases to make them work for larger and more modern models.

**Q4: Can you provide an explanation on how the formulas presented in the text were constructed?**

Based on our theoretical results we found it easy to choose coefficients in the formulas intuitively as relative strength. The examples provided throughout our paper (e.g., Table 5) were all constructed by relying on this interpretation without requiring finetuning of the constants. Intuitively chosen constants typically align well with the intended outcome, without requiring extensive tweaking. For example, while the coefficient in front of the classifier term (e.g., Table 5) might seem unintuitive, it ensures that the relative strength of the classifier is $1$ (as in Fudge). Prompts were not changed after initialization and specifically chosen to be simple, since we want to avoid extensive prompt tuning in our setting, to better demonstrate the benefits of model arithmetic at a foundational level.

---

### Meta-Review · Area_Chair_YLtW · 2023-12-11

**Metareview:**

The submission introduces a new method for controlled text generation through 'language model arithmetic', in which different models and classifiers are combined to give fine-grained control over generation. Reviewers found the approach to be novel, elegant and well motivated. It's great that the paper gives both theoretical motivations and more practical compatibility with speculative sampling for efficient inference. The main limitation is the relatively narrow scope of the evaluation, which focuses on toxicity and (in the revision) sentiment. Overall this is a creative and interesting paper that I recommend for acceptance.

**Justification For Why Not Higher Score:**

Limited evaluation

**Justification For Why Not Lower Score:**

I think some people will find this pretty interesting, so might be worth a spotlight

---

### Decision · Program_Chairs · 2024-01-16

Accept (spotlight)